# UNCERTAINTY QUANTIFICATION USING A CODEBOOK OF ENCODERS

## ABSTRACT

Many machine learning applications are limited not by the accuracy of current models but by the inability of these models to assign confidence to their predictions – the models don't know what they don't know. Among methods that do provide uncertainty estimates, there remains a tradeoff between reliable yet expensive methods (e.g., deep ensembles) and lightweight alternatives that can be miscalibrated. In this paper, we propose a lightweight uncertainty quantification method with performance comparable to deep ensembles across a range of tasks and metrics. The key idea behind our approach is to revise and augment prior information bottleneck methods with a codebook to obtain a compressed representation of all inputs seen during training. Uncertainty over a new example can then be quantified by its distance from this codebook. The resulting method, the Uncertainty Aware Information Bottleneck (UA-IB), requires only a single forward pass to provide uncertainty estimates. Our experiments show that UA-IB can achieve better Out-of-Distribution (OOD) detection and calibration than prior methods, including those based on the standard information bottleneck.

## 1 INTRODUCTION

As deep learning models move out of the lab into the real world and start impacting human lives (Larson et al., 2019; Gupta et al., 2021; Jumper et al., 2021; Jiménez-Luna et al., 2020; Roy et al., 2022; Band et al., 2021), uncertainty quantification is becoming an increasingly important problem of machine learning. This is more evident in high-stakes and safety-critical decision-making scenarios where mistakes can have severe impacts such as medical diagnosis (Roy et al., 2022; Band et al., 2021) or autonomous vehicles (Gupta et al., 2021). In these cases, model uncertainty can help users decide when to trust the model or to defer to a human expert (Kivlichan et al., 2021). Moreover, uncertainty can be used to guide exploration in reinforcement learning (Osband et al., 2021; Mai et al., 2022; Lee et al., 2021) or to query datapoints in active learning (Houlsby et al., 2011; Thrun, 1995; Nguyen et al., 2022).

Several methods have been developed that endow deep neural networks (DNNs) with an estimate of uncertainty. These methods include probabilistic approaches tailored to DNNs (Blundell et al., 2015; Osawa et al., 2019; Gal & Ghahramani, 2016) or ensemble approaches (Lakshminarayanan et al., 2017; Wilson & Izmailov, 2020). A shared characteristic of these methods is that they require multiple model samples to produce a reliable uncertainty estimate. More recent approaches aim at a deterministic characterization of uncertainty that can be computed with a single forward pass of a single model. These *Deterministic Uncertainty Methods* (DUMs) quantify uncertainty by computing a distance score or measure of a new test example from previously trained-upon datapoints. Most competitive DUMs can be classified as Gaussian Process models (Liu et al., 2020; van Amersfoort et al., 2021) or cluster-based approaches (Van Amersfoort et al., 2020). Existing DUMs rely on additional regularization techniques to obtain representations capable of identifying inputs that lie far from the support of the training dataset. Although these techniques help DUMs reach state-of-the-art out-of-distribution (OOD) detection results, they may undermine their calibration, i.e., how well DNNs can predict their incorrectness (Postels et al., 2022). Similarly, the recent work of Zhu et al. (2023) shows that OOD methods based on outlier exposure have poor misclassification prediction performance.

In this work, we introduce a rate-distortion theoretic view of uncertainty quantification in deep neural networks. Under this lens, we seek to improve the quality of uncertainty estimates using a single-model, deterministic characterization. Fig. 1 provides an overview of our approach. The key contributions of this paper are as follows:

- We formulate uncertainty quantification as the computation of a rate-distortion function to obtain a compressed representation of the training dataset. This representation is a set of prototypes defined as centroids of the training datapoints with respect to a distance measure. The expected distance of a datapoint from the centroids estimates model's uncertainty for the datapoint.

- We take a "meta-probabilistic" perspective to the rate-distortion problem. In particular, the distortion function operates on distributions of embeddings and corresponds to a statistical distance. To do so, we use the Information Bottleneck framework. The proposed formulation jointly regularizes DNN's representations and renders it uncertainty-aware.

- We design and qualitatively verify a practical deep learning algorithm that identifies the centroids of the training data.

- We show experimentally that the proposed uncertainty score outperforms other DUMs when used for OOD tasks.

- Finally, our method can detect both OOD samples and misclassified samples. In particular, the proposed uncertainty score closes the gap between DUMs and expensive ensemble methods in terms of calibration.

## 2 PRELIMINARIES

### 2.1 INFORMATION BOTTLENECK

The Information Bottleneck (IB) (Tishby et al., 2000) provides an information-theoretic view for balancing the complexity of a stochastic encoder $Z$ for input $X$ [1] and its predictive capacity for the desired output $Y$. The IB objective is:

$$\min_{\boldsymbol{\theta}} \quad -I(Z, Y; \boldsymbol{\theta}) + \beta \, I(Z, X; \boldsymbol{\theta}), \tag{1}$$

where $\beta \geq 0$ is the trade-off factor between the accuracy term $I(Z, Y; \boldsymbol{\theta})$ and the complexity term $I(Z, X; \boldsymbol{\theta})$. $\boldsymbol{\theta}$ denotes the parameters of the distributional family of encoder $p(\boldsymbol{z} \mid \boldsymbol{x}; \boldsymbol{\theta})$ [2]. In words, training by Eq. 1 encourages the model to find a representation $Z$ that is maximally expressive about output $Y$ while being maximally compressive about input $X$.

Typically, the mutual information terms in Eq. 1 cannot be computed in closed-form since they involve an intractable integration over $p(\boldsymbol{x})$ (see Eq. 20, 22 in Appendix A). The Variational Information Bottleneck (VIB) (Alemi et al., 2017) considers parametric approximations $m(\boldsymbol{y} \mid \boldsymbol{z}; \boldsymbol{\theta}), q(\boldsymbol{z}; \boldsymbol{\phi})$ of decoder $p(\boldsymbol{y} \mid \boldsymbol{z})$ and marginal distribution $p(\boldsymbol{z})$ belonging to a distributional family parametrized by $\boldsymbol{\theta}$ [3] and $\boldsymbol{\phi}$ respectively. The VIB objective maximizes a lower bound of $I(Z, Y; \boldsymbol{\theta})$ and minimizes an upper bound of $I(Z, X; \boldsymbol{\theta})$ (see Eq. 23). In this work, we reconsider the complexity term. An upper bound of this term is an expected Kullback-Leibler divergence:

$$I(Z, X; \boldsymbol{\theta}) = \mathbb{E}_X [D_{\mathrm{KL}}(p(\boldsymbol{z} \mid \boldsymbol{x}; \boldsymbol{\theta}), p(\boldsymbol{z}))] \leq \mathbb{E}_X [D_{\mathrm{KL}}(p(\boldsymbol{z} \mid \boldsymbol{x}; \boldsymbol{\theta}), q(\boldsymbol{z}; \boldsymbol{\phi}))]. \tag{2}$$

The expectation in Eq. 2 is taken, in practice, with respect to the empirical distribution of the training dataset $\mathcal{D}_{\mathrm{train}} = \{(\mathbf{x}_i, \mathbf{y}_i)\}_{i=1}^N$:

$$I(Z, X; \boldsymbol{\theta}) \lesssapprox \frac{1}{N} \sum_{i=1}^N D_{\mathrm{KL}}(p(\boldsymbol{z} \mid \boldsymbol{x}_i; \boldsymbol{\theta}), q(\boldsymbol{z}; \boldsymbol{\phi})). \tag{3}$$

### 2.2 RATE DISTORTION THEORY

The rate-distortion theory (Berger, 1971; Berger & Gibson, 1998; Cover, 1999) quantifies the fundamental limit of data compression, i.e., at least how many bits are needed to quantize data coming from a stochastic source

---

[1] We denote random variables as $X, Y, Z$ and their instances as $\boldsymbol{x}, \boldsymbol{y}, \boldsymbol{z}$.

[2] $\boldsymbol{\theta}$ will represent a function implemented by a neural network. For input $\boldsymbol{x}$ it computes the parameters of the conditional distribution $p(\cdot \mid \boldsymbol{x}; \boldsymbol{\theta})$ in its output. For example, for a Gaussian with diagonal covariance: $\boldsymbol{\theta}(\boldsymbol{x}) = \{\mu(\boldsymbol{x}), \sigma(\boldsymbol{x})\}, \mu(\boldsymbol{x}) \in \mathbb{R}^d, \sigma(\boldsymbol{x}) \in \mathbb{R}_{\geq 0}^d$. Optimization with respect to $\boldsymbol{\theta}$ will refer to optimization with respect to the weights of network $\boldsymbol{\theta}$.

[3] In the rest of the paper, we use $\boldsymbol{\theta}$ to denote the joint set of parameters of encoder and variational decoder.

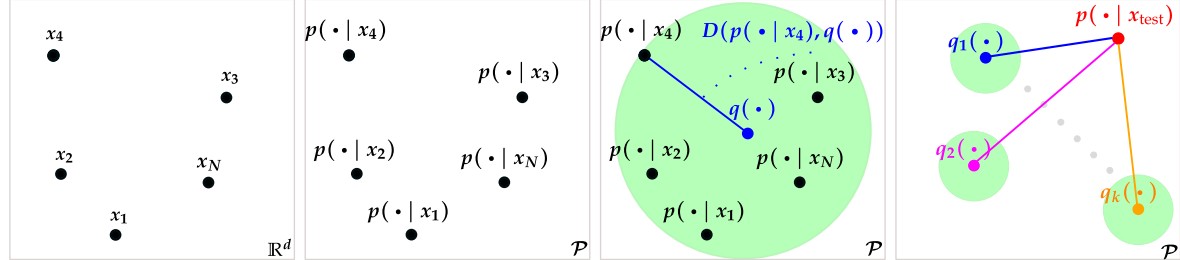

(a) $\mathcal{D}_{\text{train}}$ as a set of points in the Euclidean space $\mathbb{R}^d$.

(b) $\mathcal{D}_{\text{train}}$ as a set of points in distribution space $\mathcal{P}$.

(c) Support of $\mathcal{D}_{\text{train}}$ as a statistical ball ($k=1$).

(d) Distance from the codebook of encoders ($k>1$).

Figure 1: **Overview of UA-IB.** Uncertainty quantification in UA-IB is based on compressing the training dataset $\mathcal{D}_{\text{train}}$ by learning a codebook and computing distances from the codebook. The datapoints in $\mathcal{D}_{\text{train}}$, originally lying in $\mathbb{R}^d$ (1a), are embedded into distribution space $\mathcal{P}$ of a parametric family of distributions through their encoders (1b). Compression of $\mathcal{D}_{\text{train}}$ amounts to finding the centroids of the encoders in terms of a statistical distance $D$ (1c). For complex datasets, usually multiple centroids are needed (1d). The uncertainty for a previously unseen test datapoint is quantified by its expected distance from the codebook: $uncertainty(x_{\text{test}}) = \mathbb{E}[D(p(z \mid x_{\text{test}}; \theta), q_\kappa(z; \phi))]$.

given a desired fidelity. Formally, consider random variable $X \sim p(x)$ with support set[4] $\mathcal{X}$. Data coming from source $X$ will be compressed by mapping them to a random variable $\hat{X}$ with support set $\hat{\mathcal{X}}$. It is common to refer to $\hat{X}$ as the source code or quantization of $X$. In this work, we consider a discrete source over $\mathcal{D}_{\text{train}}$ following the empirical distribution. The formal description is deferred to Section 4.1.

The quality of the reconstructed data is assessed using a distortion function $D : \mathcal{X} \times \hat{\mathcal{X}} \to \mathbb{R}^+$. The rate-distortion function $R(D^*)$ is defined as the minimum achievable rate (number of bits) of the quantization scheme for a prescribed level of expected distortion $D^*$:

$$R(D^*) \triangleq \min_{p(\hat{x}|x)} \quad I(X; \hat{X}) \quad \text{subject to} \quad \mathbb{E}_{X, \hat{X}}[D(x, \hat{x})] \leq D^*. \tag{4}$$

It can be shown that $R(D^*)$ in Eq. 4 is equivalent to a double minimization problem over $p(\hat{x})$, $p(\hat{x} \mid x)$ (Lemma 10.8.1 of Cover (1999)). This equivalence enables an alternating minimization algorithm (Csiszár, 1984) – the Blahut–Arimoto (BA) algorithm (Blahut, 1972; Matz & Duhamel, 2004) – for computing $R(D^*)$. In practice, numerical computation of the rate-distortion function through the BA algorithm is often infeasible, primarily due to lack of knowledge of the optimal support of $\hat{X}$. The *Rate Distortion Fixed Cardinality (RDFC)* formulation (Rose, 1994; Banerjee et al., 2004) simplifies the computation of $R(D^*)$ by assuming finite support $\hat{\mathcal{X}}$ that is jointly optimized:

$$R^k(D^*) \triangleq \min_{\hat{\mathcal{X}}, p(\hat{x}|x)} \quad I(X; \hat{X}) \quad \text{subject to} \quad \mathbb{E}_{X, \hat{X}}[D(x, \hat{x})] \leq D^*, \quad |\hat{\mathcal{X}}| = k. \tag{5}$$

By taking the Lagrangian formulation of Eq. 5 for the distortion constraint, we get:

$$\min_{\hat{\mathcal{X}}, p(\hat{x}|x)} \quad I(X, \hat{X}) + \alpha \mathbb{E}_{\hat{X}, X}[D(x, \hat{x})] \text{ subject to } |\hat{\mathcal{X}}| = k, \tag{6}$$

where $\alpha \geq 0$ is the optimal Lagrange multiplier that corresponds to the distortion tolerance $D^{*}$[5]. The RDFC objective in Eq. 6 can be greedily estimated by alternating optimization over $\hat{\mathcal{X}}$, $p(\hat{x})$, $p(\hat{x} \mid x)$ yielding a solution that is locally optimal (Banerjee et al., 2004).

## 3 MOTIVATION

The idea behind our approach is visualized in Fig. 1. The crux of our approach is the observation that the variational marginal $q(z; \phi)$ in Eq. 2 and Eq. 3 encapsulates all encoders $p(z \mid x_i; \theta)$ of datapoints in $\mathcal{D}_{\text{train}}$

---

[4]Support set of $X \sim p(x)$ is the set $\mathcal{X} = \{x : p(x) > 0\}$.

[5]Formally, $\alpha$ is a function of $D^*$: $\alpha \equiv \alpha(D^*)$. However, we omit this dependence for notational brevity.

encountered during training. To see this formally, we introduce a random variable $P_X$ defined by $X \sim p(\boldsymbol{x})$. The value of $P_X$ corresponding to $\boldsymbol{x}$ is the encoder's density $p(\boldsymbol{z} \mid \boldsymbol{x}; \boldsymbol{\theta})$ (see Fig. 1a, 1b). In other words, a value of $P_X$ is itself a probability distribution. From proposition 1 of Banerjee et al. (2005), $\mathbb{E}[P_X]$ is the unique centroid of encoders $p(\boldsymbol{z} \mid \boldsymbol{x}_i; \boldsymbol{\theta})$ with respect to *any* Bregman divergence $D_f$ defined by a strictly convex and differentiable function $f$ (Bregman, 1967; Brekelmans & Nielsen, 2022) (see def. A.1):

$$\mathbb{E}[P_X] = \frac{1}{N} \sum_{i=1}^{N} p(\boldsymbol{z} \mid \boldsymbol{x}_i; \boldsymbol{\theta}) = \arg \min_{q(\boldsymbol{z})} \frac{1}{N} \sum_{i=1}^{N} D_f(p(\boldsymbol{z} \mid \boldsymbol{x}_i; \boldsymbol{\theta}), q(\boldsymbol{z})). \quad (7)$$

We note that the upper bound in Eq. 3 emerges as a special case of the minimization objective in Eq. 7. This is because the Kullback-Leibler divergence is a Bregman divergence (Azoury & Warmuth, 2001; Nielsen et al., 2007) with the negative entropy as generator function (Frigyik et al., 2008; Csiszár, 1995)[6]. Therefore, $q(\boldsymbol{z}; \boldsymbol{\phi})$ in the VIB can also be viewed as a variational centroid of the training datapoints' encoders (see Fig. 1c). In this work, we consider learnable parameters $\boldsymbol{\phi}$. Under this view, the role of the regularization term $I(Z, X; \boldsymbol{\theta})$ when upper bounded by Eq. 2 is now twofold: i) it both regularizes encoder $p(\boldsymbol{z} \mid \boldsymbol{x}; \boldsymbol{\theta})$ and ii) it learns a distributional centroid $q(\boldsymbol{z}; \boldsymbol{\phi})$ for encoders $p(\boldsymbol{z} \mid \boldsymbol{x}_i; \boldsymbol{\theta})$ of training examples $\boldsymbol{x}_i$.

For complex data, it usually does not suffice to represent $\{p(\boldsymbol{z} \mid \boldsymbol{x}_i; \boldsymbol{\theta})\}_{i=1}^{N}$ by a single distribution $q(\boldsymbol{z}; \boldsymbol{\phi})$. Therefore, we will need to learn a collection of $k$ centroids $\{q_\kappa(\boldsymbol{z}; \boldsymbol{\phi})\}_{\kappa=1}^{k}$[7] (see Fig. 1d). Moreover, standard Euclidean distances might omit aspects of data that are essential for characterizing distance from $\mathcal{D}_{\text{train}}$. In Section 4, we formalize how such a set of distributions can be learned and used to quantify distance from $\mathcal{D}_{\text{train}}$. In Section 5, we empirically confirm our hypothesis.

## 4 UNCERTAINTY AWARE INFORMATION BOTTLENECK

### 4.1 MODEL

In this section, we present *Uncertainty Aware Information Bottleneck (UA-IB)*: An IB problem with a complexity constraint that regularizes the network *and* renders the network distance-aware given a compressed representation of $\mathcal{D}_{\text{train}}$. Distance from this representation helps the network decide whether it has sufficient evidence for reaching a reliable prediction. We keep an information-geometric interpretation of this representation. We argue that an input $\boldsymbol{x}$ is better characterized by its encoder $p(\boldsymbol{z} \mid \boldsymbol{x}; \boldsymbol{\theta})$. In this case, the features of $\boldsymbol{x}$ and the codes used for computing distance from $\mathcal{D}_{\text{train}}$ lie in the parameter space of a distributional family $\mathcal{P}$ [8].

The mathematical construction of our work was alluded in Section 3 when we introduced random variable $P_X$. $P_X$ is defined by $X$ and takes as value the distribution $p(\boldsymbol{z} \mid \boldsymbol{x}; \boldsymbol{\theta})$, i.e., the encoder, as we sample $X \sim p(\boldsymbol{x})$. In its empirical form over a finite number of $N$ training datapoints $\mathcal{D}_{\text{train}}$, the distribution of $P_X$ is a discrete *distribution over distributions*: $P_X$ is discrete taking values in the set $\mathcal{P}_X = \{p(\boldsymbol{z} \mid \boldsymbol{x}_i; \boldsymbol{\theta})\}_{i=1}^{N}$ with probability $1/N$. We also define a random variable $Q$. By fixing the number $k$ of distributional centroids, $Q$ takes values $[q_1(\boldsymbol{z}; \boldsymbol{\phi}), q_2(\boldsymbol{z}; \boldsymbol{\phi}), \ldots, q_k(\boldsymbol{z}; \boldsymbol{\phi})]$ following distribution $\pi = [\pi(1), \pi(2), \ldots, \pi(k)]$. Its support set is denoted by $\mathcal{Q}$. $\pi_{\boldsymbol{x}}$ is the conditional assignment probabilities of encoder $p(\boldsymbol{z} \mid \boldsymbol{x}; \boldsymbol{\theta})$ to the centroids such that $\pi_{\boldsymbol{x}} = [\pi_{\boldsymbol{x}}(1), \pi_{\boldsymbol{x}}(2), \ldots, \pi_{\boldsymbol{x}}(k)]$ with:

$$\pi_{\boldsymbol{x}}(\kappa) = p(Q = q_\kappa(\boldsymbol{z}; \boldsymbol{\phi}) \mid P_X = p(\boldsymbol{z} \mid \boldsymbol{x}; \boldsymbol{\theta})). \quad (8)$$

Compression of $\mathcal{D}_{\text{train}}$ is phrased as a RDFC problem (see Eq. 5) for the source of encoders $P_X$ using the source code $Q$:

$$R^k(D^*; \boldsymbol{\theta}) = \min_{\mathcal{Q}, \pi_{\boldsymbol{x}}} \quad I(P_X, Q; \boldsymbol{\theta}, \boldsymbol{\phi}) \quad \text{subject to} \quad \mathbb{E}_{P_X, Q}[D(p(\boldsymbol{z} \mid \boldsymbol{x}; \boldsymbol{\theta}), q_\kappa(\boldsymbol{z}; \boldsymbol{\phi}))] \leq D^*, \quad |\mathcal{Q}| = k. \quad (9)$$

At this point, we underline that the source of encoders $P_X$ depends on $\boldsymbol{\theta}$. Since centroids $q_\kappa(\boldsymbol{z}; \boldsymbol{\phi})$ are used to quantize the set of encoders in $\mathcal{D}_{\text{train}}$, we will also call them code distributions. We will refer to $\mathcal{Q} = \{q_\kappa(\boldsymbol{z}; \boldsymbol{\phi})\}_{\kappa=1}^{k}$ as the codebook. Albeit in this work we investigate only the behavior of the Kullback-Leibler divergence, the distortion function $D$ in Eq. 9 can be *any* statistical distance measure between two *probability distributions*. To get a better insight into Eq. 9, we consider two edge cases. In the case of a single code, i.e.,

---

[6]We consider functional Bregman divergences, i.e., the generalization of Bregman divergence that acts on functions.

[7]$\boldsymbol{\phi}$ will represent the joint set of the parameters of all centroids $q_\kappa$.

[8]For example, for the family $\mathcal{P} = \{p(\boldsymbol{z}; \boldsymbol{\theta}) \mid \boldsymbol{\theta} \in \boldsymbol{\Theta}\}$ of $d-$ dimensional Normal distributions, the parameter space is $\boldsymbol{\Theta} = \{\mathbb{R}^d \times \text{Sym}^+(d; \mathbb{R})\}$ with $\text{Sym}^+(d; \mathbb{R})$ the set of $d \times d$ symmetric, positive definite matrices.

$k = 1$, with $D$ taken as the Kullback-Leibler divergence, Eq. 3 is equivalent to the VIB (Alemi et al., 2017) (upper bound in Eq. 3). For $k = N$, the optimal codes would correspond to training datapoints' encoders: $q_\kappa(z; \phi) = p(z \mid x_\kappa; \theta)$ yielding zero compression (and regularization).

Optimizing with respect to the support set $\mathcal{Q}$ amounts to optimizing with respect to parameters $\phi$ of codes $q_\kappa(z; \phi)$. Therefore, the problem in Eq. 9 can be written as:

$$R^k(D^*; \theta) = \min_{\phi, \pi_x} \quad I(P_X, Q; \theta, \phi) \quad \text{subject to} \quad \mathbb{E}_{P_X, Q}[D(p(z \mid x; \theta), q_\kappa(z; \phi))] \leq D^*. \quad (10)$$

The Uncertainty Aware IB (UA-IB) replaces the rate term $I(Z, X; \theta)$ of the IB ( Eq. 1) with the achievable rate $R^k(D^*; \theta)$ of Eq. 10 given distortion tolerance $D^*$. Formally, the UA-IB of cardinality $k$ is defined as:

$$\min_{\theta} \quad -I(Z, Y; \theta) + \beta R^k(D^*; \theta) \iff \min_{\theta} \min_{\phi, \pi_x} \quad \mathcal{L},$$

$$\text{where } \mathcal{L} \triangleq -I(Z, Y; \theta) + \beta I(P_X, Q; \theta, \phi) + \alpha \beta \, \mathbb{E}_{P_X, Q}[D(p(z \mid x; \theta), q_\kappa(z; \phi))]. \quad (11)$$

The equivalence in Eq. 11 stems from the Lagrangian form of the RDFC (see Eq. 6). Training the network with the loss function $\mathcal{L}$ encourages encoders $p(z \mid x; \theta)$ whose samples $z$ are informative about output $y$ while staying statistically close to codes $q_\kappa(z; \phi)$. As in the standard VIB (Alemi et al., 2017), $I(Z, Y; \theta)$ can be estimated by the lower bound $\mathbb{E}_{X,Z}[\log m(y \mid z; \theta)]$ that is maximized with respect to variational decoder $m(y \mid z; \theta)$ (see Eq. 23).

## 4.2 Learning Algorithm

The optimization problem of Eq. 11 can be solved by alternating minimizations (Banerjee et al., 2004). We note that $I(P_X, Q; \theta, \phi)$ in Eq. 11 is tractable since $P_X, Q$ are discrete random variables taking $N$ (size of training dataset) and $k$ (size of codebook) possible values, respectively. At each step, a single block of parameters is optimized. The most recent value is used for the parameters that are not optimized at the step. The internal minimization step corresponds to the computation of RDFC (Eq. 6). The minimization steps are summarized as follows:

$$\text{repeat} \begin{cases} t. & \text{Update decoder } m(y \mid z; \theta), \text{encoder } p(z \mid x; \theta): \quad \theta \leftarrow \theta - \eta_\theta \nabla_\theta \mathcal{L} \\ t+1. & \text{Update conditional assignment probabilities } \pi_x \text{ (from Eq. 12)} \\ t+2. & \text{Update centroids } q_\kappa(z; \phi): \quad \phi \leftarrow \phi - \eta_\phi \nabla_\phi \mathcal{L} \\ t+3. & \text{Update marginal assignment probabilities } \pi \text{ (from Eq. 13)} \end{cases}$$

Steps $t + 1$, $t + 3$ are computationally cheap and can be performed analytically with a single forward pass:

$$\pi_x(\kappa) = \frac{\pi(\kappa)}{\mathcal{Z}_x(\alpha)} \exp\left(-\alpha D(p(z \mid x; \theta), q_\kappa(z; \phi))\right), \quad (12) \qquad \pi(\kappa) = \frac{1}{N} \sum_{i=1}^{N} \pi_{x_i}(\kappa), \quad (13)$$

where $\mathcal{Z}_x(\alpha)$ is the partition function: $\mathcal{Z}_x(\alpha) = \sum_{\kappa=1}^{k} \pi(\kappa) \exp\left(-\alpha D(p(z \mid x; \theta), q_\kappa(z; \phi))\right)$. $\pi_x$ in Eq. 12 (see also Eq. 10.124 of Cover (1999)) assigns higher probability to the centroid statistically closer in terms of $D$ to the encoder of $x$. $\pi$ in Eq. 13 is derived in Lemma 10.8.1 of Cover (1999) and is the marginal of $\pi_x$.

Steps $t, t+2$ require back-propagation and correspond to gradient descent steps. The pseudocode of our method (Algorithm 1) along with a practical implementation for mini-batch training is given in Appendix B.

## 4.3 Uncertainty Quantification in the IB

The solution to the problem of Eq. 11 provides us with codes $q_\kappa(z; \phi)$ for encoders in $\mathcal{D}_{\text{train}}$ (see Fig. 1d). Large distance from these codes signals an unfamiliar input $x$ for which the network should be less confident when predicting $y$. Formally, we define uncertainty over datapoint $x$ as the conditional expected distortion:

$$\text{uncertainty}(x) = \text{distance}(x, \mathcal{D}_{\text{train}}) = \mathbb{E}_{Q|P_X = p(z|x;\theta)}\left[D(p(z \mid x; \theta), q_\kappa(z; \phi))\right]. \quad (14)$$

The distribution of $Q$ in the expectation of Eq. 14 conditioned on encoder $p(z \mid x; \theta)$ (also defined in Eq. 8) is given in Eq. 12. The expectation in Eq. 14 is taken over a finite number of values $k$. Therefore, the uncertainty score of Eq. 14 can be computed *deterministically* with a *single forward pass* of the network without requiring Monte Carlo approximations.

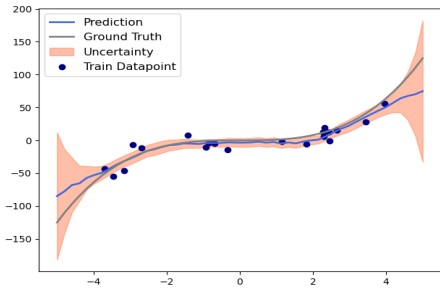

(a) A single cluster of training data points.

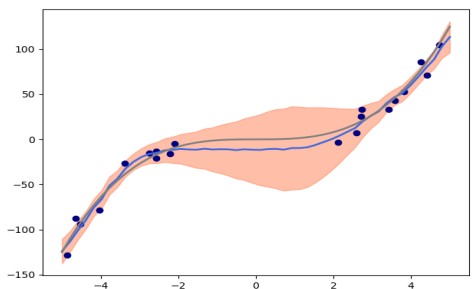

(b) Two clusters of training data points.

Figure 2: **Uncertainty estimation on noisy regression tasks.** We consider the Kullback-Leibler divergence as the distortion function in the uncertainty score of Eq. 14. A larger distance from the training datapoints (blue dots) is consistently quantified by higher uncertainty (width of pink area). Moreover, the true values lie well within $\pm 2 \times$ the proposed uncertainty score around the predictive mean.

## 5 EXPERIMENTS

### 5.1 SYNTHETIC EXAMPLE

Before we compare with other DUMs, we first need to sanity-check the proposed model and learning algorithm. Synthetic experiments are handy for this task since they allow us to control the aspects of the problem and test the behavior of the model under different conditions. In this work, we apply UA-IB to synthetic regression tasks. In Fig. 2, we visualize the predictive uncertainty, i.e., the value of the distortion function in Eq. 14, for two cases of training datasets. We provide details for training the models in Appendix D.2.1.

In Fig. 2a, we first follow the original set-up of Hernández-Lobato & Adams (2015). We generate 20 training data points from the uniform distribution $\mathcal{U}[-4, 4]$. The test data are evenly taken in $[-5, 5]$. The targets are $y = x^3 + \epsilon$, where $\epsilon \sim \mathcal{N}(0, 9)$. We use a single centroid to represent the whole dataset. We verify that as we move far away from the data, the model's confidence and accuracy decline.

In Fig. 2b, we stress test our model under a harder variant of the first problem. In this case, we create two clusters of training data points sampled from $\mathcal{U}[-5, -2]$ and $\mathcal{U}[2, 5]$. We use two codes. Wilson & Izmailov (2020) show that Fig. 2b corresponds to a typical failure case of many uncertainty-aware methods since they end up being overconfident in the area between the clusters of the training datapoints. However, UA-IB faithfully exhibits higher uncertainty in this area.

### 5.2 UNCERTAINTY AWARE IB FOR OUT-OF-DISTRIBUTION DETECTION

All approaches do not require auxiliary OOD datasets either to train the models or to tune hyperparameters. We compare UA-IB against a deterministic baseline, an ensemble baseline, a VIB with a mixture marginal trained with gradient descent, and the most competitive deterministic, single forward pass uncertainty methods. For all the methods, we use Wide ResNet 28-10 (Zagoruyko & Komodakis, 2016) as backbone architecture. For UA-IB and VIB, we use 8-dimensional latent features. For UA-IB, we consider $k = 10$ centroids. For VIB, we consider a mixture with 10 components. We use the Kullback-Leibler divergence as the distortion function in Eq. 14. For fair comparisons, we train the IB and the Gaussian Process models with a single latent sample. Further training and evaluation configurations are given in Appendix D.2.2.

To compare the uncertainty quality between different models, we evaluate their performance in distinguishing between the test sets of CIFAR-10 and OOD datasets using each model's uncertainty score. The OOD datasets we consider are SVHN (Netzer et al., 2019) and CIFAR-100. As shown in Table 1, UA-IB outperforms all baselines in terms of AUPRC and AUROC (the positive label in the binary OOD classification corresponds to the OOD images). We confirm that distances at a distributional granularity are more informative compared to Euclidean distances. The accuracy of our model is on par with that of the baselines (see Table 2). Moreover, the model can become uncertainty-aware without substantially increasing the model size (see Table 2). The additional parameters correspond to centroids' parameters that are implemented as trainable constant tensors.

Table 1: **OOD performance of baselines trained on the CIFAR-10 dataset.** We consider two tasks: a standard OOD task for distinguishing SVHN (OOD dataset) from CIFAR-10, and a harder one for distinguishing CIFAR-100 (OOD dataset) from CIFAR-10. In bold are top results (within standard error), and the horizontal lines separate ensembles and dataset augmentation methods from single forward pass methods. The performance of all models is averaged over 10 random seeds. UA-IB outperforms all baselines in all tasks with respect to all metrics.

| Method | SVHN | | CIFAR-100 | |
|---|---|---|---|---|
| | AUROC ↑ | AUPRC ↑ | AUROC ↑ | AUPRC ↑ |
| Deep Ensemble of 5 (Lakshminarayanan et al., 2017) | $0.97 \pm 0.004$ | $0.984 \pm 0.003$ | $0.916 \pm 0.001$ | $0.902 \pm 0.002$ |
| Packed Ensembles of 4 (Laurent et al., 2023) | 0.97 | 0.98 | NA | NA |
| Masked Ensembles (Durasov et al., 2021) | 0.94 | 0.96 | NA | NA |
| RegMixup (Pinto et al., 2022) | 0.967 | NA | 0.896 | NA |
| Deterministic (Zagoruyko & Komodakis, 2016) | $0.956 \pm 0.004$ | $0.976 \pm 0.004$ | $0.892 \pm 0.002$ | $0.88 \pm 0.002$ |
| DUQ (Van Amersfoort et al., 2020) | $0.940 \pm 0.003$ | $0.956 \pm 0.006$ | $0.817 \pm 0.012$ | $0.826 \pm 0.006$ |
| DUE (van Amersfoort et al., 2021) | $0.958 \pm 0.005$ | $0.968 \pm 0.015$ | $0.871 \pm 0.011$ | $0.865 \pm 0.011$ |
| SNGP (Liu et al., 2020) | $0.971 \pm 0.003$ | $0.987 \pm 0.001$ | $0.908 \pm 0.003$ | $0.907 \pm 0.002$ |
| vanilla VIB (Alemi et al., 2018) | $0.715 \pm 0.081$ | $0.869 \pm 0.039$ | $0.663 \pm 0.045$ | $0.701 \pm 0.034$ |
| **UA-IB (ours)** | $\mathbf{0.986 \pm 0.004}$ | $\mathbf{0.994 \pm 0.002}$ | $\mathbf{0.922 \pm 0.002}$ | $\mathbf{0.915 \pm 0.002}$ |

Table 2: **Accuracy and model size of OOD baselines**. Although we use a narrow bottleneck (8-dimensional latent variables), the accuracy of our model is not compromised compared to other deterministic uncertainty baselines. This is because 10 distributional codes can sufficiently represent the training dataset without diminishing the regularization effect and distance awareness of the rate-distortion constraint. More importantly, UA-IB can inject uncertainty awareness into the model with a very small model size overhead.

| Method | Accuracy ↑ | # Trainable Parameters ↓ |
|---|---|---|
| Deep Ensemble of 5 (Lakshminarayanan et al., 2017) | **96.6%** | $182,395,970$ |
| Deterministic (Zagoruyko & Komodakis, 2016) | 96.2% | **36,479,194** |
| DUQ (Van Amersfoort et al., 2020) | 94.9% | 40,568,784 |
| DUE (van Amersfoort et al., 2021) | 95.6% | 36,480,314 |
| SNGP (Liu et al., 2020) | 95.9% | 36,483,024 |
| vanilla VIB (Alemi et al., 2018) | 95.9% | 36,501,042 |
| UA-IB (ours) | 95.9% | 36,501,114 |

Figure 3: **Qualitative evaluation of encoders' codebook.** We visualize the number of test data points per class assigned to each centroid during training. We assign a data point to the centroid with the smallest statistical distance from its encoder. Each centroid progressively attracts data points of the same class. Moreover, all centroids are assigned a non-zero number of test datapoints. Therefore, the centroids are useful for better explaining both train and previously unseen, test data points.

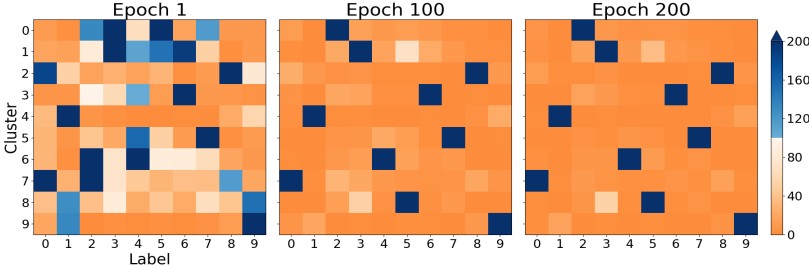

To qualitatively inspect the learning algorithm of Section 4.2, in Fig. 3 we plot the number of test datapoints per class represented by each centroid $q_\kappa(z; \phi)$ in different training phases. We note that the class counts refer to the *true* and not the *predicted* class. As training proceeds, UA-IB learns similar latent features for datapoints that belong to the same class and pushes them closer to the same centroid. Certain centroids, however, conflate

test datapoints of different classes. For example, a small number of test datapoints of class 3 (cat) are assigned to (are closest to) the centroid whose majority class is 5 (dog). Assignment to the wrong centroid presages model's misprediction for these datapoints. In contrast, we observed in a similar figure on the training datapoints that they are perfectly separated by the centroids (there are only blue squares in the colormap).

We ablate the UA-IB hyperparameters in Appendix D.3, and demonstrate that UA-IB can solve OOD on regression problems in Appendix D.1.

### 5.3 UNCERTAINTY AWARE IB FOR CALIBRATION

In Table 3, we evaluate the performance of various uncertainty scores in predicting the model's incorrectness (Corbière et al., 2019; Zhu et al., 2022). We report the *Calibration AUROC* that was introduced by Kivlichan et al. (2021) and later used by Postels et al. (2022) for evaluating calibration of deterministic uncertainty methods. As pointed out by Postels et al. (2022), the ECE (Expected Calibration Error) is not the appropriate metric for DUMs since their uncertainty scores are not directly reflected to the probabilistic forecast. In Section 7, we discuss how deterministic uncertainty scores can be leveraged during both training and evaluation for further improving the model's predictions. Another benefit of Calibration AUROC compared to ECE is that it cannot be trivially reduced using post hoc calibration heuristics such as temperature scaling (Guo et al., 2017). In contrast, Calibration AUROC focuses on the intrinsic ability of the model to distinguish its correct from incorrect predictions and the ranking performance of its uncertainty score, i.e., whether high uncertainty predictions are wrong. We remark that such property can be particularly useful in unsupervised learning tasks where, for example, the uncertainty score can guide exploration in deep reinforcement learning (Osband et al., 2016; Lee et al., 2021; Mai et al., 2022) or active learning tasks (Houlsby et al., 2011; Thrun, 1995; Nguyen et al., 2022). In Fig. 6, we provide the proposed model's calibration plot.

Table 3: **Calibration AUROC of deterministic uncertainty methods.** We examine how well the model can predict it will be wrong from its estimated uncertainty. The problem is framed as a binary classification task with the positive label indicating a mistake. UA-IB closes the gap between single forward pass models and ensembles (Postels et al., 2022).

| Method | Uncertainty Description | Calibration AUROC ↑ |
|---|---|---|
| Deep Ensemble of 5 (Lakshminarayanan et al., 2017) | Members' disagreement (variance) | $0.922 \pm 0.003$ |
| DUQ (Van Amersfoort et al., 2020) | Euclidean distance ($l_2$-norm) from centroid | $0.889 \pm 0.013$ |
| DUE (van Amersfoort et al., 2021) | Gaussian Process variance | $0.856 \pm 0.026$ |
| SNGP (Liu et al., 2020) | Dempster-Shafer uncertainty | $0.897 \pm 0.006$ |
| **UA-IB (ours)** | **Statistical distance (KL) from centroid** | $\mathbf{0.93} \pm \mathbf{0.003}$ |

## 6 RELATED WORK

**Single-Model Uncertainty Methods.** Deep ensembles (Lakshminarayanan et al., 2017) are widely regarded as a strong baseline for estimating epistemic uncertainty. Typically, they require training of multiple identical models from different initializations. Therefore, they come with high computational and memory footprint. Single-model approaches seek to improve the parameter efficiency of uncertainty estimation methods. Batchensemble (Wen et al., 2020) and its Bayesian extension (Dusenberry et al., 2020) consider an ensemble of low-rank perturbations of the weights that are trained end-to-end under a single model. Probabilistic methods such as Bayesian Neural Networks (BNNs) (Blundell et al., 2015; Osawa et al., 2019; Wenzel et al., 2020; Louizos & Welling, 2017) and Monte Carlo (MC) Dropout (Gal & Ghahramani, 2016) have also been deployed for representing uncertainty in neural networks. Albeit single-model approaches, all aforementioned methods still require multiple forward passes to estimate the network's confidence over the final prediction. Our work falls into a recently emerged line of research called *single forward pass* or *deterministic uncertainty estimation* (DUM) methods. In principle, these methods aim at explicitly computing a distance score from the training data for a test input. Gaussian Processes (GPs) are intrinsically distance-aware models since they are defined by a kernel function that quantifies similarity to the training datapoints. SNGP (Liu et al., 2020) relies on a Laplace approximation of the GP based on Random Fourier Features (RFF) (Rahimi & Recht, 2007). DUE (van Amersfoort et al., 2021) uses the inducing point GP approximation (Titsias, 2009). Both models enforce bi-Lipschitz constraints on the network by spectral normalization (Miyato et al., 2018) to encourage sensitivity and smoothness of the extracted features. More closely related to our work is DUQ (Van Amersfoort et al., 2020). Similar

to our work, DUQ quantifies uncertainty as the distance from centroids responsible for representing the training data. The distance is computed in terms of a Radial Basis Function (RBF) kernel. In contrast to our work, DUQ is trained with respect to a binary cross entropy loss function that assigns datapoints to clusters in a supervised manner. Moreover, the number of centroids is hardwired to the number of classes. This restriction renders difficult deployment of the model to regression tasks or classification tasks with a large number of classes. Prior to our work, distances in all DUMs are defined over Euclidean norms. On the other hand, in UA-IB, uncertainty is defined by statistical distances. Using distributions (more holistic objects) and the Kullback-Leibler divergence (more sensitive distance measure) are the key components that improve UA-IB's performance compared to these baselines. Our experiments do not use efficient data augmentation methods such as AugMix (Hendrycks et al., 2020) and RegMixup (Pinto et al., 2022). These methods are orthogonal to ours and they can provide a performance boost without increasing the computational cost.

**Connections with Maximum Likelihood Mixture Estimation.** Limited work has sought connections between Maximum Likelihood Mixture Estimation (MLME) and computation of the rate-distortion function. Banerjee et al. (2004) prove the equivalence between these two problems for Bregman distortions and exponential families. In this case and under the assumption of constant variance for all mixture's components, learning the support set in RDFC corresponds to learning the mixture means. For MLME on parametric distributions, i.e., encoders, a straightforward way to leverage this connection is to define the "sample space" of the MLME as the "parameter space" of encoder's distribution family[9]. Similarly, training with a mixture (for the marginal) VIB (Alemi et al., 2018) entails an MLME problem where the data points (to be clustered) are latent samples drawn from encoders. To get better insights, in Appendix C we anatomize the loss function. As we saw in Table 1, a full statistical description of encoders (instead of using a finite–single in the experiment– number of its samples) along with the proposed alternating minimization algorithm that guides assignments to centroids during training, helps UA-IB capture uncertainty *exactly* with a *single* forward pass. From a theoretical standpoint, deriving rigorous connections between the two problems would be interesting for future work.

## 7 LIMITATIONS & FUTURE WORK

The main focus of this work is to define and analyze a more comprehensive notion of distance from the training data manifold under the auspices of Information Bottleneck methods. However, the proposed distance is not utilized during training for eventually *improving* model's predictions. This is the case with other similar DUMs such as DUQ. In UA-IB, this can be achieved by designing a *stochastic* decoder that induces variance proportionate to the estimated uncertainty in its final prediction. Such a decoder can be viewed as a distance-aware epinet (Osband et al., 2021). Although in this work we used the Kullback-Leibler divergence, the proposed framework is flexible and supports inference with alternative statistical distances tailored to the application of interest (Minka, 2005; Nielsen, 2023). On a separate note, uncertainty estimates have been used in Deep reinforcement learning for safety and risk estimation or exploration (Lee et al., 2021; Osband et al., 2016). However, these methods rely on a large number of ensemble members (Lee et al., 2021) or forward passes (Wu et al., 2021) to obtain credible estimates. Moreover, it has been empirically found that predictive variance-based uncertainty scores are inadequate in off-policy DRL tasks (Appendix D.2. in (Fujimoto et al., 2018)). Application of DUMs, and UA-IB in particular, to unsupervised learning settings is another promising research area. Finally, we remark that UA-IB, similar to VIB (Alemi et al., 2017), can be applied post-hoc to a large, pretrained backbone network without requiring interference with its expensive training process. As such, post-hoc integration of UA-IB can render it suitable for industrial machine learning applications.

## 8 CONCLUSION

We introduced UA-IB, a deterministic uncertainty method for deep neural networks. We framed uncertainty quantification as a rate-distortion problem to learn a lossy compression of the training dataset. We used the Information Bottleneck formalism where the compression (regularization) term acts on the encoder's density function and the distortion function corresponds to a statistical distance. We designed a practical learning algorithm that is based on successive refinement of the rate-distortion function estimates. Experimental analysis shows that our method outperforms baselines in OOD tasks. Moreover, the proposed uncertainty score correlates better with model's performance and improves its calibration.

---

[9]For example, clustering Gaussian encoders could be phrased as MLME of means and symmetric positive definite covariance matrices with a mixture of Normal-Wishart distributions.

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

## A  PRELIMINARIES

**Definition A.1** (***Bregman Divergence***).  Let $f : \mathcal{S} \to \mathbb{R}$ be a differentiable, strictly convex function of Legendre type on a convex set $\mathcal{S} \subseteq \mathbb{R}^d$. The *Bregman divergence* $D_f$ for any two points $\boldsymbol{x}, \boldsymbol{y} \in S$ is defined as (Bregman, 1967):

$$D_f(\boldsymbol{x}, \boldsymbol{y}) = f(\boldsymbol{x}) - f(\boldsymbol{y}) - \langle \boldsymbol{x} - \boldsymbol{y}, \nabla f(\boldsymbol{y}) \rangle, \tag{15}$$

where $\nabla f(\boldsymbol{y}) \in \mathbb{R}^d$ denotes the gradient of $f$ evaluated at $\boldsymbol{y}$.

**Definition A.2** (***Dual Bregman Form of Exponential Family***).  A multivariate parametric family of distributions $\mathcal{F}_\psi = \{p(\cdot; \boldsymbol{\phi}) \mid \boldsymbol{\phi} \in \Phi\}$ with $\Phi = \mathrm{int}(\Phi) = \mathrm{dom}(\psi) \subseteq \mathbb{R}^p$ is called regular exponential family if each probability density in the family is of the form:

$$p(\boldsymbol{x}; \boldsymbol{\phi}) = \exp(\langle t(\boldsymbol{x}), \boldsymbol{\phi} \rangle - \psi(\boldsymbol{\phi})) \, h_0(\boldsymbol{x}), \ \ \forall \boldsymbol{x} \in \mathbb{R}^d, \tag{16}$$

where $t(\boldsymbol{x})$ is a minimal sufficient statistic for the family. $\boldsymbol{\phi}$ is called the natural parameter. $\psi(\boldsymbol{\phi})$ is the log-partition function that normalizes the density function. It can be shown (Barndorff-Nielsen, 2014) that $\Phi$ is a non-empty convex set in $\mathbb{R}^d$ and that $\psi$ is a convex function. $h_0(\boldsymbol{x})$ is a base measure that does not depend on $\boldsymbol{\phi}$.

From Theorem 4 by Banerjee et al. (2005), the density of Eq. 16 can be written as:

$$p(\boldsymbol{x}; \boldsymbol{\phi}) = \exp\left(-D_{\psi^*}(t(\boldsymbol{x}), \hat{\boldsymbol{t}}(\boldsymbol{\phi}))\right) f_{\psi^*}(\boldsymbol{x}), \tag{17a}$$

where $\psi^*$ is the convex conjugate of $\psi$, $\hat{\boldsymbol{t}}$ is the expectation of the sufficient statistic and :

$$f_{\psi^*}(\boldsymbol{x}) = \exp\left(\psi^*(t(\boldsymbol{x}))\right) h_0(\boldsymbol{x}), \tag{17b}$$

$$\hat{\boldsymbol{t}} \triangleq \mathbb{E}_X[t(\boldsymbol{x})] \equiv \hat{\boldsymbol{t}}(\boldsymbol{\phi}) = \nabla \psi(\boldsymbol{\phi}). \tag{17c}$$

From the definition of $f_{\psi^*}(\boldsymbol{x})$ in Eq. 17b, we highlight that it is a function that does not depend on $\boldsymbol{\phi}$. Therefore, when we train by Maximum Likelihood Estimation (MLE) to learn $\boldsymbol{\phi}$, this term can be omitted from the objective function. The equality in Eq. 17c is derived by differentiating $\int p(\boldsymbol{x}; \boldsymbol{\phi}) d\boldsymbol{x} = 1$ with respect to $\boldsymbol{\phi}$ and by making use of Eq. 16 and the definition of $\hat{\boldsymbol{t}}$.

Eq. 17a is called the *Bregman form* of the density in Eq. 16.

**Definition A.3** (***Scaled Exponential Family***).  Given an exponential family $\mathcal{F}_\psi$ with natural parameter $\boldsymbol{\phi}$ and log-partition function $\psi(\boldsymbol{\phi})$ (see Eq. 16), a *scaled exponential family* (Jiang et al., 2012) $\mathcal{F}_\psi^\alpha$ with $\alpha > 0$ has natural parameter $\tilde{\boldsymbol{\phi}} = \alpha\boldsymbol{\phi}$ and log-partition function $\tilde{\psi}(\tilde{\boldsymbol{\phi}}) = \alpha\psi(\tilde{\boldsymbol{\phi}}/\alpha) = \alpha\psi(\boldsymbol{\phi})$. Its Bregman form is (Jiang et al., 2012):

$$p(\boldsymbol{x}; \tilde{\boldsymbol{\phi}}) = \exp\left(-\alpha D_{\psi^*}(t(\boldsymbol{x}), \hat{\boldsymbol{t}}(\boldsymbol{\phi}))\right) f_{\alpha\psi^*}(\boldsymbol{x}), \tag{18}$$

where $\psi^*$ is the convex conjugate of $\psi$. $f_{\alpha\psi^*}$ is defined in Eq. 17b where we scale $\psi^*$ by $\alpha$. Finally, the mean $\hat{\boldsymbol{t}}$ of $\mathcal{F}_\psi^\alpha$ is the same with that of $\mathcal{F}_\psi$ and is given in Eq. 17c.

**Variational Information Bottleneck**.  Alemi et al. (2017) derive efficient variational estimates of the mutual information terms in Eq. 1. The accuracy term is:

$$I(Z, Y; \boldsymbol{\theta}) = \int \log \frac{p(\boldsymbol{y} \mid \boldsymbol{z})}{p(\boldsymbol{y})} p(\boldsymbol{y}, \boldsymbol{z}) d\boldsymbol{z} d\boldsymbol{y}. \tag{19}$$

However, Eq. 19 cannot be computed analytically since the decoder $p(\boldsymbol{y} \mid \boldsymbol{z})$ requires an intractable marginalization with respect to $\boldsymbol{x}$:

$$p(\boldsymbol{y} \mid \boldsymbol{z}) = \int \frac{p(\boldsymbol{y}|\boldsymbol{x})p(\boldsymbol{z}|\boldsymbol{x}; \boldsymbol{\theta})p(\boldsymbol{x})}{p(\boldsymbol{z})} d\boldsymbol{x}. \tag{20}$$

Similarly, the regularization term:

$$I(Z, X; \boldsymbol{\theta}) = \int \log \frac{p(\boldsymbol{z} \mid \boldsymbol{x}; \boldsymbol{\theta})}{p(\boldsymbol{z})} p(\boldsymbol{x}, \boldsymbol{z}) d\boldsymbol{z} d\boldsymbol{x} \tag{21}$$

contains the intractable marginal:

$$p(\boldsymbol{z}) = \int p(\boldsymbol{z} \mid \boldsymbol{x}; \boldsymbol{\theta}) p(\boldsymbol{x}) d\boldsymbol{x}. \tag{22}$$

Variational estimates in a distributional family $m(\boldsymbol{y} \mid \boldsymbol{z}; \boldsymbol{\theta})$ and $q(\boldsymbol{z}; \boldsymbol{\phi})$ of Eq. 20, Eq. 22 minimize the Kullback-Leibler divergences $D_{\mathrm{KL}}(m(\boldsymbol{y} \mid \boldsymbol{z}; \boldsymbol{\theta}), p(\boldsymbol{y} \mid \boldsymbol{z}))$ and $D_{\mathrm{KL}}(q(\boldsymbol{z}; \boldsymbol{\phi}), p(\boldsymbol{z}))$, respectively. Non-negativity of the Kullback-Leibler divergence yields a lower bound of Eq. 19 and an upper bound of Eq. 21. The resulting Variational Information Bottleneck (VIB) loss is:

$$\mathcal{L}_{\mathrm{VIB}} = \mathbb{E}_{X,Z}[-\log m(\boldsymbol{y} \mid \boldsymbol{z}; \boldsymbol{\theta})] + \beta \mathbb{E}_X[D_{\mathrm{KL}}(p(\boldsymbol{z} \mid \boldsymbol{x}; \boldsymbol{\theta}), q(\boldsymbol{z}; \boldsymbol{\phi}))]. \tag{23}$$

## B  LEARNING ALGORITHM ( SECTION 4.2 CONTINUED.)

---

**Algorithm 1:** Optimization of Uncertainty Aware IB

---

**Inputs:**
  training data: $\mathcal{D}_{\mathrm{train}} = \{(\mathbf{x}_i, \mathbf{y}_i)\}_{i=1}^N$
  codebook size: $k$
  statistical distance: $D$
  hyper-parameters:
    temperature $\alpha \geq 0$ (see Eq. 12)
    regularization coefficient $\beta \geq 0$ (see Eq. 11)
**Outputs:**
  optimal parameters of encoder and decoder: $\boldsymbol{\theta}^\dagger$
  optimal codebook parameters: $\boldsymbol{\phi}^\dagger$
  marginal assignment probabilities: $\pi^\dagger$
**Initialize:**
  encoder $p(\boldsymbol{z} \mid \boldsymbol{x}; \boldsymbol{\theta})$
  decoder $m(\boldsymbol{y} \mid \boldsymbol{z}; \boldsymbol{\theta})$
  codebook $\{q_\kappa(\boldsymbol{z}; \boldsymbol{\phi})\}_{\kappa=1}^k$
  $\pi, \pi_{\boldsymbol{x}_i}$ to uniform distribution
**while** *not converged* **do**
  Update decoder $m(\boldsymbol{y} \mid \boldsymbol{z}; \boldsymbol{\theta})$, encoder $p(\boldsymbol{z} \mid \boldsymbol{x}; \boldsymbol{\theta})$: $\boldsymbol{\theta} \leftarrow \boldsymbol{\theta} - \eta_{\boldsymbol{\theta}} \nabla_{\boldsymbol{\theta}} \mathcal{L}$ ($\mathcal{L}$ in Eq. 11)
  **for** $i = 1, 2, \ldots, N$ **do**
    **for** $\kappa = 1, 2, \ldots, k$ **do**
      $\pi_{\boldsymbol{x}_i}(\kappa) = \frac{\pi(\kappa)}{\mathcal{Z}_{\boldsymbol{x}_i}(\alpha)} \exp\left(-\alpha D(p(\boldsymbol{z} \mid \boldsymbol{x}; \boldsymbol{\theta}), q_\kappa(\boldsymbol{z}; \boldsymbol{\phi}))\right)$ (Eq. 12)
    **end**
  **end**
  Update codes $q_\kappa(\boldsymbol{z}; \boldsymbol{\phi})$: $\boldsymbol{\phi} \leftarrow \boldsymbol{\phi} - \eta_{\boldsymbol{\phi}} \nabla_{\boldsymbol{\phi}} \mathcal{L}$ ($\mathcal{L}$ in Eq. 11)
  **for** $\kappa = 1, 2, \ldots, k$ **do**
    $\pi(\kappa) = \frac{1}{N} \sum_{i=1}^N \pi_{\boldsymbol{x}_i}(\kappa)$ (Eq. 13)
  **end**
**end**

---

The gradient descent updates in Algorithm 1 are written using constant learning rates $\eta_{\boldsymbol{\theta}}, \eta_{\boldsymbol{\phi}}$. In practice, we can use any optimizer with adaptive learning rates. To make sure that the conditional assignment probabilities $\pi_{\boldsymbol{x}}$ are kept fixed, we apply a `tf.stop_gradient` operator when the gradients $\nabla_{\boldsymbol{\theta}} \mathcal{L}, \nabla_{\boldsymbol{\phi}} \mathcal{L}$ are computed. To render the update of $\pi$ amenable to mini-batch optimization, we maintain a moving average such that at batch $t$ of size $B$:

$$\pi^0(\kappa) = 1/k, \ \pi^t(\kappa) = \gamma \pi^{t-1}(\kappa) + (1 - \gamma) \frac{1}{B} \sum_{i=1}^B \pi_{\boldsymbol{x}_i}(\kappa). \tag{24}$$

$0 \leq \gamma \leq 1$ is the momentum of the moving average. On the onset of an epoch, $\pi$ is set to the moving average of the previous epoch. Similarly, the moving average is reset to the uniform distribution.

In this work, we use multivariate Gaussian distributions for centroids and encoders. In this case, the centroids' parameters $\phi$ correspond to the means and covariance matrices: $\phi = \{\boldsymbol{\mu}_\kappa, \boldsymbol{\Sigma}_\kappa\}_{\kappa=1}^k$ and the optimal solution has a closed form (Davis & Dhillon, 2006). We empirically observed that using the closed-form update for the covariance matrix and the gradient descent for the means facilitates optimization and speeds-up convergence.

In all experiments, we use a full-covariance multivariate Gaussian for the encoder and the centroids. The encoder's network first learns a matrix $\boldsymbol{S}$ as $\boldsymbol{S} = \boldsymbol{U}\sqrt{\boldsymbol{\Lambda}}$. $\boldsymbol{U}$ is a unitary matrix. $\boldsymbol{\Lambda}$ is a positive definite, diagonal matrix. $\boldsymbol{U}$ and $\boldsymbol{\Lambda}$ are computed from the SVD decomposition of a symmetric matrix. To enforce positive definiteness of $\boldsymbol{\Lambda}$ with small initial values, we transform its entries by $\texttt{softplus}(\lambda - 5.0)$. A similar transformation was used by Alemi et al. (2017). Finally, the covariance matrix is given by: $\boldsymbol{\Sigma} = \boldsymbol{S}\boldsymbol{S}^T$.

## C  VIB FOR EUCLIDEAN CLUSTERING OF LATENT CODES

One way we can use the set of distributions $\{q_\kappa(\boldsymbol{z}; \phi)\}_{\kappa=1}^k$ is to consider a mixture of $k$ distributions for the marginal $q(\boldsymbol{z}; \phi)$ and trivially train it by gradient descent (Alemi et al., 2018). To better understand the role of each $q_\kappa(\boldsymbol{z}; \phi)$ during optimization, we associate a discrete random variable $\hat{Z}$ with $Z$. The value of $\hat{Z}$ indicates the assignment of $Z$ to a component $q_\kappa(\boldsymbol{z}; \phi)$ of the mixture. We rewrite the upper bound of Eq. 2 in terms of $\hat{Z}$. The resulting decomposition of Proposition 1 shows that the regularization term in the VIB encloses the objective of a fixed-cardinality rate-distortion function (see Eq. 6) under some assumptions. However, computation of Eq. 25 requires Monte-Carlo samples of $Z$ to assign an encoder to the mixture components. The regularization terms of VIB and UA-IB are identical for $k = 1$. The rate-distortion formulation of Eq. 25 motivates the proposed Information Bottleneck objective of Section 4.1. It also serves as a conceptual step towards the definition of a rate-distortion function acting directly on probability densities.

**Proposition 1.** *Let the variational marginal $q(\boldsymbol{z}; \phi)$ of Eq. 2 be a mixture of $k$ distributions in $\mathbb{R}^d$ that belong to the scaled regular exponential family (see def. A.3) $\mathcal{F}_\psi^\alpha$ with $\alpha > 0$ and log-partition function $\psi$. Let $\hat{\boldsymbol{t}}_\kappa$ be the expected value of the minimal sufficient statistic $\boldsymbol{t}(Z)$ of the family when $Z \sim q_\kappa(\boldsymbol{z}; \phi)$. Let $\hat{Z}$ be a (latent) categorical random variable following distribution $\pi = [\pi(\kappa)]_{\kappa=1}^k$. We assume $\hat{Z}$ is conditionally independent of $X$ given $Z$, i.e., $P(X, Y, Z) = P(X)P(Z \mid X)P(\hat{Z} \mid Z)$. The upper bound of the VIB in Eq. 2 can be decomposed as:*

$$\mathbb{E}_X[D_{KL}(p(\boldsymbol{z} \mid \boldsymbol{x}; \boldsymbol{\theta}), q(\boldsymbol{z}; \phi))] =$$
$$- H(Z \mid X; \boldsymbol{\theta}) - \mathbb{E}_{X,Z}[\log f_{\psi^*}(\boldsymbol{z})] + \alpha\mathbb{E}_{X,Z,\hat{Z}}[D_{\psi^*}(\boldsymbol{t}(\boldsymbol{z}), \hat{\boldsymbol{t}}_{\hat{z}}(\phi))] + I(Z, \hat{Z} \mid X; \boldsymbol{\theta}, \phi), \quad (25)$$

*where $D_{\psi^*}$ is the Bregman divergence of $\mathcal{F}_\psi$, i.e., the Bregman divergence defined by the convex conjugate function $\psi^*$ of $\psi$. $f_{\psi^*} : dom(\psi^*) \to \mathbb{R}^+$ is a uniquely defined function that does not depend on the natural parameter $\phi$.*

*Proof.* We expand Eq. 2:

$$\mathbb{E}_X[D_{KL}(p(\boldsymbol{z} \mid \boldsymbol{x}; \boldsymbol{\theta}), q(\boldsymbol{z}; \phi))] =$$
$$\int p(\boldsymbol{x})p(\boldsymbol{z} \mid \boldsymbol{x}; \boldsymbol{\theta}) \log p(\boldsymbol{z} \mid \boldsymbol{x}; \boldsymbol{\theta})d\boldsymbol{z}d\boldsymbol{x} - \int p(\boldsymbol{x})p(\boldsymbol{z} \mid \boldsymbol{x}; \boldsymbol{\theta}) \log q(\boldsymbol{z}; \phi)d\boldsymbol{z}d\boldsymbol{x}. \quad (26)$$

The first term of Eq. 26 is the negative conditional differential entropy of the encoder, i.e., $-H(Z \mid X; \boldsymbol{\theta})$. We will focus on the second term of Eq. 26. For a fixed $\boldsymbol{z}$:

$$\log q(\boldsymbol{z}; \phi) = \mathbb{E}_{\hat{Z}|\boldsymbol{z}}[\log q(\boldsymbol{z}; \phi)]$$
$$= \mathbb{E}_{\hat{Z}|\boldsymbol{z}}[\log q(\boldsymbol{z}; \phi) + \log p(\hat{\boldsymbol{z}} \mid \boldsymbol{z}; \phi) - \log p(\hat{\boldsymbol{z}} \mid \boldsymbol{z}; \phi)]$$
$$= \mathbb{E}_{\hat{Z}|\boldsymbol{z}}[\log q(\boldsymbol{z}, \hat{\boldsymbol{z}}; \phi) - \log p(\hat{\boldsymbol{z}} \mid \boldsymbol{z}; \phi)]$$
$$= \mathbb{E}_{\hat{Z}|\boldsymbol{z}}[\log q(\boldsymbol{z} \mid \hat{\boldsymbol{z}}; \phi) + \log p(\hat{\boldsymbol{z}}) - \log p(\hat{\boldsymbol{z}} \mid \boldsymbol{z}; \phi)]. \quad (27)$$

By definition of $\hat{Z}$, $P(\hat{Z} = \kappa) = \pi(\kappa)$ and $q(\boldsymbol{z} \mid \kappa; \phi) = q_\kappa(\boldsymbol{z}; \phi)$. Let $\hat{\boldsymbol{t}}_\kappa(\phi)$ be the expected value of $\boldsymbol{t}(Z)$ when $Z \sim q_\kappa(\boldsymbol{z}; \phi)$. Since $q_\kappa(\boldsymbol{z}; \phi)$ belongs to a class of the exponential family, its Bregman form (definition A.2) is:

$$q_\kappa(\boldsymbol{z}; \phi) = \exp\left(-D_{\psi^*}(\boldsymbol{t}(\boldsymbol{z}), \hat{\boldsymbol{t}}_\kappa(\phi))\right) f_{\psi^*}(\boldsymbol{z}), \quad (28)$$

where $\psi^*$ is the convex conjugate of the log-partition function $\psi$ of the family, $D_{\psi^*}$ is the Bregman divergence defined by $\psi^*$, and $f_{\psi^*} : \text{dom}(\psi^*) \to \mathbb{R}^+$ is a uniquely defined function that does not depend on $\phi$, i.e., the natural parameter of $q_\kappa$. In general, we can consider a scaled exponential family with Bregman form (see definition A.3):

$$q_\kappa(\boldsymbol{z}; \boldsymbol{\phi}) = \exp\left(-\alpha D_{\psi^*}(\boldsymbol{t}(\boldsymbol{z}), \hat{\boldsymbol{t}}_\kappa(\boldsymbol{\phi}))\right) f_{\alpha\psi^*}(\boldsymbol{z}), \; \alpha > 0. \tag{29}$$

By taking expectation of Eq. 27 with respect to $p(\boldsymbol{x}), p(\boldsymbol{z} \mid \boldsymbol{x}; \boldsymbol{\theta})$ and using Eq. 29, we can rewrite Eq. 26:

$$\mathbb{E}_X[D_{KL}(p(\boldsymbol{z} \mid \boldsymbol{x}; \boldsymbol{\theta}), q(\boldsymbol{z}; \boldsymbol{\phi}))] =$$
$$- H(Z \mid X; \boldsymbol{\theta}) - \mathbb{E}_{X,Z}[\log f_{\psi^*}(\boldsymbol{z})] + \alpha \mathbb{E}_{X,Z,\hat{Z}}[D_{\psi^*}(\boldsymbol{t}(\boldsymbol{z}), \hat{\boldsymbol{t}}_{\hat{z}}(\boldsymbol{\phi}))] + I(Z, \hat{Z} \mid X; \boldsymbol{\theta}, \boldsymbol{\phi}). \tag{30}$$

$\square$

By minimizing Eq. 25 with respect to $\boldsymbol{\theta}$, encoder $p(\boldsymbol{z} \mid \boldsymbol{x}; \boldsymbol{\theta})$ is regularized to generate samples $\boldsymbol{z}$ whose sufficient statistics are close to the means $\hat{\boldsymbol{t}}_\kappa$ in terms of $D_\psi^*$. For Gaussian $q_\kappa(\boldsymbol{z}; \boldsymbol{\phi})$, minimizing Eq. 25 would encourage small covariance for the encoder and mixture's components. This is counterbalanced by the negative entropy term in Eq. 25, without which the encoder would collapse to a single atom, i.e., to one of the expected values of $q_\kappa(\boldsymbol{z}; \boldsymbol{\phi})$. Maximizing its entropy by the first term in Eq. 25 avoids such degenerate solutions. The scale $\alpha$ balances the trade-off between the distortion level and the stochasticity of $Z$. A similar observation following, however, an entirely algebraic route, is also made by Hoffman et al. (2017). Here, we present an information-theoretic perspective of this trade-off.

By minimizing Eq. 25 with respect to $\boldsymbol{\phi}$, $q(\boldsymbol{z}; \boldsymbol{\phi})$ is trained to maximize the expected log-likelihood $\mathbb{E}_{Z|X}[\log q(\boldsymbol{z}; \boldsymbol{\phi})]$ when $Z \sim p(\boldsymbol{z} \mid \boldsymbol{x}; \boldsymbol{\theta})$. Keeping everything but $\boldsymbol{\phi}$ fixed, minimizing Eq. 25 is equivalent to minimizing only the last two terms in Eq. 25 since $H(Z \mid X; \boldsymbol{\theta})$ and $\log f_{\psi^*}(\boldsymbol{z})$ do not depend on $\boldsymbol{\phi}$. When the source equals the empirical distribution over a finite number of sampled latent codes, minimizing Eq. 25 is equivalent to MLE with a mixture distribution. In case of a Gaussian mixture, in turn, it is equivalent to soft K-means clustering in latent space $\mathbb{R}^d$. For distributions for which $\boldsymbol{t}(\boldsymbol{z}) = \boldsymbol{z}$[10], minimizing Eq. 25 with respect to $\boldsymbol{\phi}$ amounts to computing the Rate Distortion Finite Cardinality (RDFC) function (Eq. 5) with the Bregman distortion $D_{\psi^*}$. The support $\hat{\mathcal{Z}}$ of $\hat{Z}$ to be learned has cardinality $k$ and corresponds to the sufficient statistic means $\hat{\mathcal{Z}} = \{\hat{\boldsymbol{t}}_\kappa\}_{\kappa=1}^k$[11]. A similar equivalence between RDFC and MLE with mixtures in the exponential family is derived by Banerjee et al. (2004). Note that in our case, the log-likelihood of latent codes sampled by the encoder is maximized instead. Moreover, the source (encoder) is not known beforehand but its parameters $\boldsymbol{\theta}$ are trainable during optimization.

---

[10] For example, for Gaussian $\mathcal{N}(\boldsymbol{\mu}, \boldsymbol{\Sigma})$ with constant and a priori known covariance matrix $\boldsymbol{\Sigma}$, $\boldsymbol{t}(\boldsymbol{z}) = \boldsymbol{z}$, $\boldsymbol{\phi} = \boldsymbol{\mu} = \hat{\boldsymbol{t}}$.

[11] We implicitly redefine $\hat{Z}$ to take values in $\hat{\mathcal{Z}} = \{\hat{\boldsymbol{t}}_\kappa\}_{\kappa=1}^k$.

# D  EXPERIMENTS (CONTINUED.)

## D.1  OUT-OF-DISTRIBUTION DETECTION ON REGRESSION TASKS

Currently, the bulk of uncertainty-aware methods is designed for and applied to image classification in supervised learning settings. However, as shown by Jaeger et al. (2023), a wide range of tasks and datasets should be considered when evaluating OOD methods. Moreover, there is an ongoing importance to effective uncertainty estimation for regression tasks, especially in unsupervised learning scenarios. For example, in deep reinforcement learning, uncertainty quantification for the estimated Q-values can be leveraged for efficient exploration (Lee et al., 2021). As already pointed out, UA-IB provides a unified notion of uncertainty for both regression and classification tasks.

In Tables 4, 5, we test the OOD capability of our model on normalized UCI datasets when trained on the Concrete Compressive Strength and the Energy Efficiency datasets (Markelle Kelly, 1998). As in the image classification tasks, the positive label corresponds to the OOD inputs. The results were averaged across 10 runs. We contrast our model with ensemble methods. We see that UA-IB consistently demonstrates OOD capability across all datasets and OOD tasks (of various difficulty). In particular, UA-IB outperforms 2-member ensembles on all OOD tasks for both datasets and 4-member ensembles for the energy efficiency dataset on all OOD tasks. In Section D.2.3, we provide the details of the experimental setup. Here, we comment that all centroids were assigned a roughly equal number of datapoints indicating the need for codebook sizes larger than one (recall that UA-IB with a unit-size codebook corresponds to the standard VIB (Alemi et al., 2017)).

Table 4: OOD performance of UA-IB trained on the UCI concrete strength dataset.

| OOD Dataset | Model | OOD Scores | |
|---|---|---|---|
| | | AUROC ↑ | AUPRC ↑ |
| kin8nm | UA-IB | $0.858 \pm 0.025$ | $0.98 \ \pm 0.004$ |
| | Ensemble of 2 | $0.791 \pm 0.017$ | $0.969 \pm 0.003$ |
| | Ensemble of 4 | $0.893 \pm 0.014$ | $0.98 \ \pm 0.002$ |
| energy efficiency | UA-IB | $0.768 \pm 0.054$ | $0.776 \pm 0.048$ |
| | Ensemble of 2 | $0.725 \pm 0.038$ | $0.721 \pm 0.049$ |
| | Ensemble of 4 | $0.854 \pm 0.028$ | $0.845 \pm 0.036$ |
| protein structure | UA-IB | $0.905 \pm 0.028$ | $0.998 \pm 0.001$ |
| | Ensemble of 2 | $0.797 \pm 0.051$ | $0.994 \pm 0.002$ |
| | Ensemble of 4 | $0.886 \pm 0.027$ | $0.997 \pm 0.001$ |
| boston housing | UA-IB | $0.821 \pm 0.049$ | $0.752 \pm 0.068$ |
| | Ensemble of 2 | $0.745 \pm 0.038$ | $0.686 \pm 0.039$ |
| | Ensemble of 4 | $0.836 \pm 0.049$ | $0.792 \pm 0.049$ |

Table 5: OOD performance of UA-IB trained on the UCI energy efficiency dataset.

| OOD Dataset | Model | OOD Scores | |
|---|---|---|---|
| | | AUROC ↑ | AUPRC ↑ |
| kin8nm | UA-IB | $0.982 \pm 0.008$ | $0.998 \pm 0.001$ |
| | Ensemble of 2 | $0.916 \pm 0.025$ | $0.992 \pm 0.003$ |
| | Ensemble of 4 | $0.977 \pm 0.008$ | $0.998 \pm 0.001$ |
| concrete strength | UA-IB | $0.978 \pm 0.011$ | $0.988 \pm 0.006$ |
| | Ensemble of 2 | $0.898 \pm 0.043$ | $0.941 \pm 0.028$ |
| | Ensemble of 4 | $0.967 \pm 0.02$ | $0.979 \pm 0.013$ |
| protein structure | UA-IB | $0.989 \pm 0.017$ | $1.0 \quad \pm 0.001$ |
| | Ensemble of 2 | $0.875 \pm 0.059$ | $0.998 \pm 0.001$ |
| | Ensemble of 4 | $0.971 \pm 0.018$ | $0.999 \pm 0.001$ |
| boston housing | UA-IB | $0.988 \pm 0.008$ | $0.988 \pm 0.007$ |
| | Ensemble of 2 | $0.888 \pm 0.043$ | $0.887 \pm 0.047$ |
| | Ensemble of 4 | $0.969 \pm 0.028$ | $0.967 \pm 0.03$ |

## D.2 IMPLEMENTATION DETAILS

### D.2.1 IMPLEMENTATION DETAILS FOR SECTION 5.1

We use a network with 3 dense layers. We apply the Uncertainty Aware IB to the last one. The intermediate layers have 100 hidden units and ELU non-linearity. We perform 1000 training iterations. We use a single encoder sample during training. The optimizer of both the main network and the centroids are set to `tf.keras.optimizers.Adam(learning_rate=1e-2)`. The rest of the hyperparameters are set to the default values of `tf.keras.keras.layers.Dense`. Regarding the initialization of encoders and centroids, we follow the setup described in Section D.2.2.

In Table 6, we provide the hyperparameters related to UA-IB. Note that the dataset for these tasks consists of only 20 datapoints. Therefore, we can use the whole dataset at each update step. In this case, there is no need to maintain moving averages for the update of the assignment probabilities and covariance matrices.

### D.2.2 IMPLEMENTATION DETAILS FOR SECTIONS 5.2, 5.3

All models are trained on four 32GB V100 GPUs.

For a fair comparison with the single forward pass methods of Table 1, we backpropagate through a single encoder sample. We apply the Information Bottleneck to the last dense layer of the classifier.

We train the means of the centroids using `tf.keras.optimizers.Adam(learning_rate=1e-1)`. Only for the case $k = 1$ in Table 10, we used `tf.keras.optimizers.Adam(learning_rate=1e-3)`.The centroid means are initialized with `tf.random_normal_initializer(mean=0.0,stddev=0.1)`. For the hyperparameters that are not related to the UA-IB, we preserve the default values used in: `https://github.com/google/uncertainty-baselines/blob/main/baselines/cifar/deterministic.py`.

Table 6: **A summary of UA-IB hyperparameters for the synthetic regression tasks.**

| Hyperparameter | Description | Value |
|---|---|---|
| $\beta$ | Regularization coefficient (see Eq. 11) | 1.0 |
| $\alpha$ | Temperature (see Eq. 12) | 5.0 |
| $\dim(\boldsymbol{z})$ | Dimension of latent features | 4 |
| $k$ | Number of centroids | 1 for Fig. 2a 2 for Fig. 2b |
| $\gamma$ | Momentum of moving averages (see Eq. 24) | 0.0 |

Table 7: **A summary of UA-IB hyperparameters for the classification tasks.**

| Hyperparameter | Description | Value |
|---|---|---|
| $\beta$ | Regularization coefficient (see Eq. 11) | 0.001 |
| $\alpha$ | Temperature (see Eq. 12) | 1.0 |
| $\dim(\boldsymbol{z})$ | Dimension of latent features | 8 |
| $k$ | Number of centroids | 10 |
| $\gamma$ | Momentum of moving averages (see Eq. 24) | 0.99 |

### D.2.3 IMPLEMENTATION DETAILS FOR SECTION D.1

For both datasets of Tables 4, 5, we used identical hyperparameters. The optimizer of the codebook was set to `tf.keras.optimizers.Adam(learning_rate=1e-1)`. The architecture consists of an MLP network with one hidden layer of dimension 50 and relu nonlinearity. The exact backbone architecture along with the hyperparameters that are not related to the UA-IB were kept the same and can be found here: `https://github.com/google/uncertainty-baselines/tree/main/baselines/uci`. Finally, we applied UA-IB between the penultimate and output layer of the architecture.

Table 8: **A summary of UA-IB hyperparameters for the UCI regression tasks.**

| Hyperparameter | Description | Value |
|---|---|---|
| $\beta$ | Regularization coefficient (see Eq. 11) | 0.001 |
| $\alpha$ | Temperature (see Eq. 12) | 1.0 |
| $\dim(\boldsymbol{z})$ | Dimension of latent features | 4 |
| $k$ | Number of centroids | 2 |
| $\gamma$ | Momentum of moving averages (see Eq. 24) | 0.99 |

### D.3 ABLATION STUDIES

In Table 9, we compare the OOD performance of UA-IB models when using other commonly-used OOD metrics. As expected, the proposed distortion score, that is *explicitly* minimized for the training datapoints via the loss function in Eq. 11, yields better OOD detection performance.

Table 9: **UA-IB performance with alternative OOD scores.** $D_{\mathrm{KL}}$ refers to the Kullback-Leibler distortion of Eq. 14. $H$ refers to the entropy of the decoder's classifier: $H \triangleq \mathbb{E}_{Y,Z|\boldsymbol{x}}[-\log m(\boldsymbol{y} \mid \boldsymbol{z}; \boldsymbol{\theta})]$. Finally, $p_{max}$ refers to the maximum probability of the classifier: $p_{\max} \triangleq \arg \max_c \mathbb{E}_{Z|\boldsymbol{x}}[m(Y = c \mid \boldsymbol{z}; \boldsymbol{\theta})]$. $p_{\max}$ and $H$ are approximated by Monte Carlo with a single sample of $Z$. The Kullback-Leibler divergence from the learned centroids is more sensitive to input variations rendering the distortion of Eq. 14 a better indicator of an OOD input.

| OOD score | SVHN | | CIFAR-100 | |
|---|---|---|---|---|
| | AUROC ↑ | AUPRC ↑ | AUROC ↑ | AUPRC ↑ |
| $D_{\mathrm{KL}}$ | **0.986 ± 0.004** | **0.994 ± 0.002** | **0.922 ± 0.002** | **0.915 ± 0.002** |
| $H$ | 0.964 ± 0.009 | 0.982 ± 0.005 | 0.891 ± 0.003 | 0.883 ± 0.003 |
| $1 - p_{max}$ | 0.959 ± 0.009 | 0.978 ± 0.006 | 0.889 ± 0.003 | 0.875 ± 0.003 |

In the rest of this section, we study the effect of the UA-IB hyperparameters, also listed in Table 6, on the OOD performance of our model.

In Table 10, we do an ablation study on cardinality $k$. We see that a larger number of centroids improves the quality of the uncertainty estimates. However, further increasing the codebook size with $k > 10$ yields diminishing performance benefits. We sought to justify this model's behavior via visual inspection of the codebook. We noticed that when $k > 10$ some centroids are assigned to only a small number of training datapoints. This observation can serve as a recipe for choosing the codebook size: albeit a larger codebook will not harm performance, unutilized entries indicate that a smaller codebook can achieve similar quality for the model's uncertainty estimates.

Table 10: **Ablation study over codebook size $k$.** A single Gaussian code $q(\boldsymbol{z})$ does not discriminate well CIFAR-10 from the visually similar datapoints of CIFAR-100. As we increase the number of centroids, UA-IB progressively becomes better at distinguishing these datasets. UA-IB reaches competitive performance with a small number of 10 centroids. The performance remains roughly the same when using a larger cardinality $k > 10$.

| | SVHN | | CIFAR-100 | |
|---|---|---|---|---|
| | AUROC ↑ | AUPRC ↑ | AUROC ↑ | AUPRC ↑ |
| $k = 10$ | **0.986 ± 0.004** | **0.994 ± 0.002** | **0.922 ± 0.002** | **0.915 ± 0.002** |
| $k = 5$ | 0.968 ± 0.031 | 0.986 ± 0.012 | 0.912 ± 0.009 | 0.907 ± 0.007 |
| $k = 1$
vanilla VIB (Alemi et al., 2017) | 0.906 ± 0.052 | 0.958 ± 0.026 | 0.746 ± 0.023 | 0.764 ± 0.026 |

In Table 11, we study the effect of the temperature $\alpha$. We verify that $\alpha$ controls the strength of the statistical distance when comparing a datapoint with the codebook. For small values of $\alpha$, the model exhibits a uniformity-tolerance for the datapoints that lie well beyond the support of the training dataset. On the other hand, the distribution $\pi_x$ of Eq. 12 becomes sharper for larger values of $\alpha$. A sharper distribution translates to a more informative centroid assignment for datapoint $x$. Subsequently, an informative codebook helps the model to successfully mark the areas of the input distribution that is familiar with.

Table 11: **Ablation study over temperature $\alpha$.** With small values of $\alpha$, the model fails to successfully discriminate inputs it should be less confident about. Large values of $\alpha$ lead to a more concentrated clustering of the training datapoints. This, in turn, provides the model with more effective OOD scores that sufficiently penalize large distances from the codebook.

|  | SVHN | | CIFAR-100 | |
|---|---|---|---|---|
|  | AUROC ↑ | AUPRC ↑ | AUROC ↑ | AUPRC ↑ |
| $\alpha = 0.1$ | $0.932 \pm 0.038$ | $0.972 \pm 0.018$ | $0.756 \pm 0.031$ | $0.776 \pm 0.032$ |
| $\alpha = 0.5$ | $0.958 \pm 0.045$ | $0.982 \pm 0.019$ | $0.878 \pm 0.057$ | $0.879 \pm 0.043$ |
| $\alpha = 1.0$ | $\mathbf{0.986 \pm 0.004}$ | $\mathbf{0.994 \pm 0.002}$ | $\mathbf{0.922 \pm 0.002}$ | $\mathbf{0.915 \pm 0.002}$ |
| $\alpha = 2.0$ | $0.989 \pm 0.003$ | $0.995 \pm 0.001$ | $0.924 \pm 0.001$ | $0.918 \pm 0.002$ |
| $\alpha = 10.0$ | $0.982 \pm 0.005$ | $0.991 \pm 0.002$ | $0.923 \pm 0.002$ | $0.916 \pm 0.002$ |

In Table 12, we vary the regularization coefficient $\beta$. We see that the model achieves the best performance within a range of $\beta$. For smaller values of $\beta$, the distortion term in Eq. 11 is disregarded. Therefore, the main network is not restricted to producing encoders that can be well-represented by the codebook. For larger values of $\beta$, the training datapoints get closely attached to the centroids. This results in statistical balls of small radius (see Fig. 1c) effectively leaving out novel, in-distribution datapoints.

Table 12: **Ablation study over regularization coefficient $\beta$.** The model is best performing for a range of values. Large values of $\beta$ correspond to small balls around the centroids (see Fig. 1c) and vice-versa. The balls should be small enough to exclude OOD inputs but large enough to include unseen, in-distribution points to which the model can generalize.

|  | SVHN | | CIFAR-100 | |
|---|---|---|---|---|
|  | AUROC ↑ | AUPRC ↑ | AUROC ↑ | AUPRC ↑ |
| $\beta = 0.0001$ | $0.925 \pm 0.429$ | $0.965 \pm 0.02$ | $0.70 \pm 0.019$ | $0.697 \pm 0.02$ |
| $\beta = 0.0005$ | $0.98 \pm 0.009$ | $0.99 \pm 0.005$ | $0.917 \pm 0.002$ | $0.91 \pm 0.003$ |
| $\beta = 0.001$ | $\mathbf{0.986 \pm 0.004}$ | $\mathbf{0.994 \pm 0.002}$ | $\mathbf{0.922 \pm 0.002}$ | $\mathbf{0.915 \pm 0.002}$ |
| $\beta = 0.005$ | $0.985 \pm 0.004$ | $0.993 \pm 0.002$ | $0.921 \pm 0.002$ | $0.914 \pm 0.002$ |
| $\beta = 0.01$ | $0.977 \pm 0.01$ | $0.988 \pm 0.005$ | $0.914 \pm 0.002$ | $0.907 \pm 0.001$ |

In Table 13, we are sweeping the bottleneck dimension. In Table 2, we see that 8-dimensional latent features can capture the information needed for the CIFAR-10 classification task. Further increasing the bottleneck size leads to irrelevant features that have no effect. On the other hand, smaller features disregard essential aspects of the input.

Table 13: **Ablation study over bottleneck dimension.** Larger latent features improve OOD capability until a performance plateaus is reached.

|  | SVHN | | CIFAR-100 | |
|---|---|---|---|---|
|  | AUROC ↑ | AUPRC ↑ | AUROC ↑ | AUPRC ↑ |
| $\dim(z) = 2$ | $0.748 \pm 0.03$ | $0.797 \pm 0.014$ | $0.678 \pm 0.014$ | $0.59 \pm 0.008$ |
| $\dim(z) = 4$ | $0.974 \pm 0.01$ | $0.98 \pm 0.004$ | $0.877 \pm 0.012$ | $0.872 \pm 0.008$ |
| $\dim(z) = 8$ | $\mathbf{0.986 \pm 0.004}$ | $\mathbf{0.994 \pm 0.002}$ | $\mathbf{0.922 \pm 0.002}$ | $\mathbf{0.915 \pm 0.002}$ |
| $\dim(z) = 10$ | $0.983 \pm 0.005$ | $0.991 \pm 0.003$ | $0.924 \pm 0.002$ | $0.915 \pm 0.001$ |

Finally, the model was not sensitive to typical values, i.e. $> 0.9$, for the momentum $\gamma$.

### D.4 QUALITATIVE EVALUATION (CONTINUED.)

In this section, we investigate qualitatively the rest of the IB methods examined in Sections 5.2 and D.3 (Tables 1, 10).

Figure 4: **Qualitative evaluation of a codebook of size 5.** We visualize the number of test data points per class assigned to each centroid at the end of three (first, middle, last) iterations of our alternating minimization algorithm (Algorithm 1). We notice that semantically similar classes are assigned to the same code. For example, dogs (class 5) and cats (class 3) are both represented by centroid 3. Similar observations hold for the pair of cars (class 1)/ trucks (class 9) and airplane (class 0)/ ships (class 8).

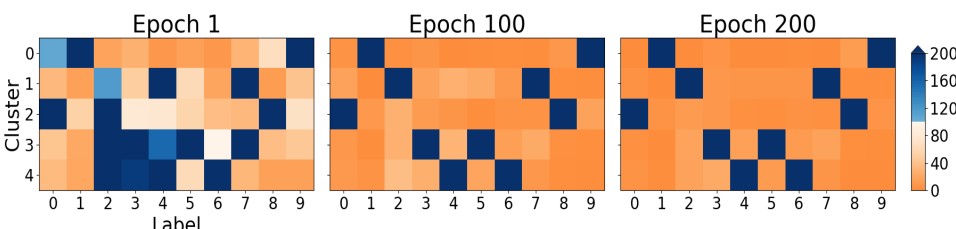

Figure 5: **Qualitative evaluation of a $10-$component mixture marginal in the VIB trained with gradient descent.** We visualize the number of test data points per class assigned to each component at the end of three (first, middle, last) epochs when the mixture variational marginal $q(\boldsymbol{z}; \boldsymbol{\phi})$ and the rest of the network (encoder and decoder) are jointly trained via gradient descent (Alemi et al., 2018). We notice that gradient descent conflates features of different classes. This observation can help explain the inferior performance of the IB gradient descent method on OOD tasks (Table 1). Moreover, it justifies the need for guiding optimization through the alternating minimization steps of Algorithm 1.

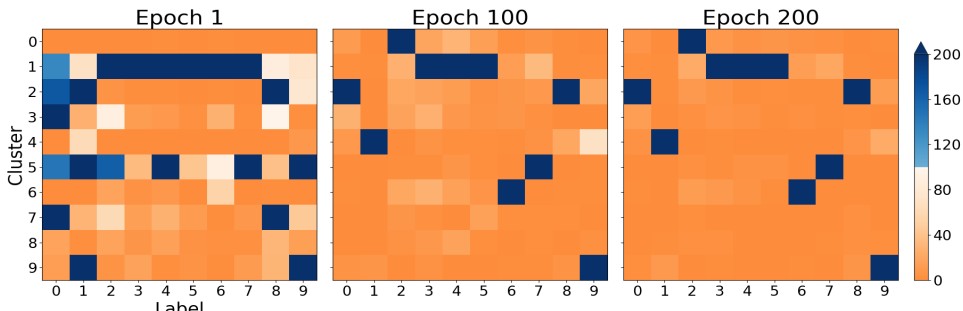

Figure 6: **Calibration plot of UA-IB on test data.** We qualitatively assess the proposed uncertainty score in terms of calibration. We train 10 models with different random seeds. For each model, we find the 20 quantiles of the estimated uncertainty on test data. We compute the accuracy for the datapoints whose uncertainty falls between two successive quantiles. We report the mean uncertainty and accuracy along with one standard deviation error bars across the runs. We see that the accuracy is higher in the quantile buckets of lower uncertainty.

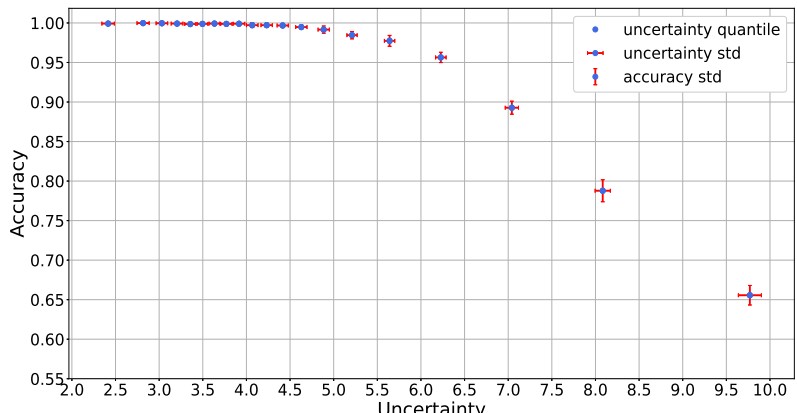

