# OpenReview forum: "Uncertainty Quantification Using a Codebook of Encoders"
_ICLR.cc/2024/Conference — Submitted to ICLR 2024_

### Official Review · Reviewer_nABa · 2023-10-26

**Soundness:** 3 good
**Presentation:** 3 good
**Contribution:** 2 fair
**Rating:** 6
**Confidence:** 4

**Summary:**

This paper develops a method for uncertainty quantification in deep networks. The proposed method uses the centroids of training distributions in a compressed feature space. It is based on the Information Bottleneck theory and it operates on a codebook of encoders. OOD experiments are on detecting images of CIFAR-10 from the CIFAR-100, and SVHN  datasets. Misclassification detection experiments are on CIFAR-10 dataset.

**Strengths:**

The approach seems sound and clear. In my view, it has the potential to have a broader impact.

Paper’s proposed method is intended to detect both OOD inputs as well as in-distribution misclassifications of the image classification models. So, one inexpensive method is proposed to perform two tasks that are usually studied separately in the literature.

Paper is well written.

I am generally positive about the paper, but there seems to be important shortcomings.

**Weaknesses:**

Experiments are thin only on small datasets. Specially, for OOD detection, the setting used in the paper is previously shown to be an insufficient indicator for generalization.

The methods used for comparison do not appear to be the most recent methods in the literature.

Literature review is weak.

-------

The reported accuracy of the models appear way below the standards in the literature. In Table 3, the ultimate accuracy of the calibrated model appears to be around 93% percent. Is that a model trained on the CIFAR-10 dataset? If yes, then this accuracy does not seem convincing. The testing accuracy of a standard ViT model on CIFAR-10 is around 98% -- with no extra computational procedure. The accuracy of the misclassification detection method by Zhu et al for WideResNet is +97%. Why would anyone use the authors’ proposed method to ultimately achieve an accuracy of 93%? Could authors extend their experiments to models with better accuracies and demonstrate that their calibration method can improve the accuracy of those models? I would suggest downloading pretrained models from the literature, e.g., the models available on HuggingFace, PyTorch library, etc.

-----

The paper below demonstrates the inadequacy of evaluating OOD detection methods on the CIFAR-10, CIFAR-100, SVHN datasets. Specifically, this reference demonstrates that when broadening the range of datasets, most of the OOD detection methods fail to generalize. Based on this, I don’t find the authors' OOD experiments convincing. I suggest authors expand their experiments to the setting used in the paper below and consider more datasets, e.g., iWildCam, BREEDS, etc.

– Jaeger, P.F., Lüth, C.T., Klein, L. and Bungert, T.J., 2022, September. A Call to Reflect on Evaluation Practices for Failure Detection in Image Classification. In The Eleventh International Conference on Learning Representations.

Experimenting on common benchmarks such as Imagenet would also be useful.

-----

I did not see any reference to the misclassification detection literature, but I might have missed it. See for example:

-- Zhu, F., Cheng, Z., Zhang, X.Y. and Liu, C.L., 2023. OpenMix: Exploring Outlier Samples for Misclassification Detection. In Proceedings of the IEEE/CVF Conference on Computer Vision and Pattern Recognition (pp. 12074-12083).

-----

Overall, because of the paper’s approach and its broad view, it appears to me that having this paper in the literature would be good, even if the proposed method does not outperform all the other competing methods. However, it would be necessary for the paper to demonstrate its capability in comparison to the most recent methods in the literature and on standard models with good accuracy, models that are freely available in the literature.

-----

I found the discussion under Limitations and Future Work frank and clear, and perhaps a better explanation of the contribution of the paper. The statement of “The main focus of this work is ….” seemed more on point to me than some of the early statements in the paper. Overall, I think introduction of the paper could be more on point, emphasizing the broad view of the paper early on.

**Questions:**

Please see questions under weaknesses.

Comparisons are made with methods that are not very recent (as summarized in tables 2 and 3). The most recent reference that authors have compared with is from 2021. Could authors please explain their choice of methods to compare with? Authors might have a reason for not considering the more recent methods of OOD detection and misclassification detection, e.g., methods from ICLR 2023 – it would be helpful to explain why.

In Figure 1, the location of points in subfigures (a) and (b) are exactly the same. Is that intentional? Paper makes a distinction between the d-dimensional Euclidean space and the distribution space denoted by P, but it is not clear to me what is the difference between these two spaces, if the depictions of points in (a) and (b) are exactly the same. What is the job of the encoder?

On the choice of k and other inputs for the method: When paper states: “For complex datasets, usually multiple centroids are needed.”, how many centroids are needed? For example, how many encoders and centroids would be needed to apply this method to models trained on Imagenet?

---

> ### Author Response · Authors · 2023-11-16
> **Response to Review**
>
> Thank you for the detailed and constructive comments! We are also glad that you think our contributions can be helpful to the community. Please find our comments below.
>
> 1) *Specially, for OOD detection, the setting used in the paper is previously shown to be an insufficient indicator for generalization.*
>
> We **extended our experiments in Appendix D.1**. We have chosen regression tasks for several reasons. First, we want to highlight that our model can handle both regression and classification tasks. Second, regression problems are often overlooked by the community. Therefore, demonstrating our model on these tasks could broaden the impact of our paper perhaps more than testing it on expensive/ larger architectures for image classification tasks.
>
> 2) *The reported accuracy of the models appear way below the standards in the literature.*
>
> **Our model does not introduce an accuracy drop.** The score 93% in Table 3 pertains to a binary classification task (where the label indicates whether the model will correctly classify the image or not) and not the image classification task. For the accuracy of the model for the image classification, please refer to Table 2.
>
> 3)  *I did not see any reference to the misclassification detection literature, but I might have missed it. See for example:*
>
> We added three references:
>
>    —In the introduction: The recent work of Zhu et al. (2023)
>
>    —In section 5.3: we evaluate the performance of various uncertainty scores in predicting the model’s incorrectness (Corbière et al., 2019; Zhu et al., 2022)
>
> 4)  *Comparisons are made with methods that are not very recent (as summarized in tables 2 and 3). The most recent reference that authors have compared with is from 2021. Could authors please explain their choice of methods to compare with?*
>
> We compared our model with single-forward pass uncertainty quantification methods that do not use image augmentations or auxiliary OOD datasets.  **We updated Table 1 to include two models published less than a year ago.**
>
> 5) *In Figure 1, the location of points in subfigures (a) and (b) are exactly the same. Is that intentional?*
>
> This is a great observation! We preserved the location of the datapoints for illustration purposes, i.e., to quickly compare x vs p( . |x). In general, we expect that the datapoints in distribution in P will be reshuffled (while the relative positions will also depend on the divergence used).
>
> 6) *Experiments are thin only on small datasets.*
>
> We would like to note that although CIFAR-10 is a smaller-scale problem, it is widely regarded as a challenging OOD task. We refer you to [1], [2]. We briefly quote here:** “CIFAR Benchmark is NOT Easier than ImageNet Benchmark “** [1]. Moreover, very impactful papers such as **SNGP and DUQ follow an experimental setup similar to ours** while they were scaled to computer vision tasks/large-scale image classification tasks later on. However, we do agree that scaling the experiments will increase the impact of the paper.
>
> 7) *For example, how many encoders and centroids would be needed to apply this method to models trained on Imagenet?*
>
> A safe estimation would be to choose a codebook size equal to the number of classes. However, smaller codebooks can still yield competitive OOD performance (see Table 7, cases k=5 vs k=10). In the revised manuscript, right above Table 10, we further discuss the codebook size selection.
>
> 8) *Overall, I think introduction of the paper could be more on point, emphasizing the broad view of the paper early on.*
>
> This is another great suggestion. We have revised the introduction:
>
> “We introduce a rate-distortion theoretic view of uncertainty quantification in deep neural networks.., Under this lens, ”
>
> References
>
> [1] Yang J, Wang P, Zou D, Zhou Z, Ding K, Peng W, Wang H, Chen G, Li B, Sun Y, Du X. Openood: Benchmarking generalized out-of-distribution detection. Advances in Neural Information Processing Systems. 2022 Dec 6;35:32598-611.
>
> [2] Van Amersfoort J, Smith L, Teh YW, Gal Y. Uncertainty estimation using a single deep deterministic neural network. InInternational conference on machine learning 2020 Nov 21 (pp. 9690-9700). PMLR.

---

> ### Comment · Reviewer_nABa · 2023-11-21
>
> I thank the authors for their detailed response.
>
> While I appreciate the improvements in the paper and the clarifications provided in the rebuttal, I still find the experiments on the CIFAR-10 dataset inadequate and not a good indicator for the usefulness of the proposed method. The additional experiments on regression tasks do not seem to go well with the CIFAR-10 experiments and adds to the confusion in my view. Hence, I keep my score.
>
> It may be worthwhile to note that evaluating OOD detection methods merely on CIFAR-10 is shown to be inadequate. See this paper which was selected as the notable top 5% of papers at ICLR2023:
>
> – Jaeger, P.F., Lüth, C.T., Klein, L. and Bungert, T.J., 2022, A Call to Reflect on Evaluation Practices for Failure Detection in Image Classification. In The Eleventh International Conference on Learning Representations.

---

> > ### Author Response · Authors · 2023-11-21
> > **response to reviewer's comments on the breadth of the datasets and tasks**
> >
> > Thank you for your reply! We would like to point out that ensemble methods are not direct competitors of our model since UAIB is leveraging a single network. In CIFAR-10 experiments in Table 1, ensembles with >>5 members would also probably outperform many of the baselines.
> >
> > Following [1], the main purpose of the UCI experiments was to demonstrate the consistent OOD capability of UAIB on regression tasks or non-image inputs.  In fact, UAIB outperforms 2-member ensembles trained on both datasets and on all OOD tasks (of various difficulty) and 4-member ensembles when trained with the energy efficiency dataset on all tasks. We agree with you and [1] that a variety of data (perhaps other than images) and tasks (perhaps other than classification) is needed. We comment on this in the updated version of the paper when motivating the experiments in section D.1: "However, as shown by Jaeger et al. (2023), a wide range of tasks and datasets should be considered when evaluating OOD methods"
> >
> > [1] Jaeger, P.F., Lüth, C.T., Klein, L. and Bungert, T.J., 2022, A Call to Reflect on Evaluation Practices for Failure Detection in Image Classification. In The Eleventh International Conference on Learning Representations.

---

> > > ### Comment · Reviewer_nABa · 2023-11-22
> > >
> > > Thank you for explaining this. I now better understand your motivation for the regression tasks -- I raise my score from 5 to 6.
> > >
> > > To be precise, my concern is that when you show your method can detect images of CIFAR-10 from SVHN, it does not necessarily imply that the method can detect the images of CIFAR-10 from other types of OOD images. This is a concern about generalization capability of your proposed method on images. It is not about how fast your method works, whether it uses ensembles of models, nor about non-image inputs.
> > >
> > > In reference [1], you can see a long list of OOD detection methods that can separate images of CIFAR-10 from SVHN, but the same models fail when it comes to separating images of CIFAR-10 from other datasets.
> > >
> > > If authors can show that their method can detect the images of CIFAR-10 from a dataset other than SVHN, e.g., iWildCam, BREEDS, or Camelyon, my concern will be addressed and I will raise my score further. This is what I suggested in my original comments.

---

> > > > ### Author Response · Authors · 2023-11-22
> > > > **on diversity of OOD tasks**
> > > >
> > > > Thank you for your clarification and the ongoing discussion!
> > > >
> > > > *"If authors can show that their method can detect the images of CIFAR-10 from a dataset other than SVHN"*
> > > >
> > > >
> > > > We refer you to Table 1, where UAIB and other DUMs are also tested on the CIFAR-100 OOD Task. Please note that by [1], CIFAR-100 is considered a near-ood dataset ( the dataset only has a semantic shift compared with the ID dataset), and hence it is more informative about the behavior and performance of the model. As shown in Table 1, the performance gain of UAIB compared to other DUMs is more pronounced for this task.
> > > >
> > > > One explanation behind UA-IB's consistency (on both cifar-10 and uci datasets and tasks) is that it's a **principled and theoretically justified method**: the OOD score of Eq. 14 increases as we move far away from the support of the training dataset (represented by the codebook). This is in contrast to methods that are empirically demonstrated and where for example one type of dataset augmentation might favor one OOD task but not the other.
> > > >
> > > > [1] Yang J, Wang P, Zou D, Zhou Z, Ding K, Peng W, Wang H, Chen G, Li B, Sun Y, Du X. Openood: Benchmarking generalized out-of-distribution detection. Advances in Neural Information Processing Systems. 2022 Dec 6;35:32598-611.

---

> ### Author Response · Authors · 2023-11-22
> **on diversity of OOD tasks (part 2)**
>
> After further inspection of the details in [1], we understand that cifar10 vs cifar100 is also a failure case of many baselines. We quote here:
>
> "DG-Res, Devries, and ConfidNet fail to generalize beyond the scenarios they have been proposed on, i.e. to more complex data sets (like iWildCam or BREEDS) and covariate distribution shifts (corruptions and sub-class shifts)."
> CIFAR-100 is characterized as a sub-class shift: "Further sub-class shifts are considered in the form of super-classes of CIFAR-100" also noted a s-ncs (semantic new class shift) in Table 1 of [1].
>
> In Table 1, this failure case is depicted as differences in the colormap of columns c100 and svhn under the cifar-10 dataset in the rows corresponding to the methods mentioned above. UA-IB passes this failure case: it consistently outperforms all methods at both tasks.
>
> [1] Jaeger, P.F., Lüth, C.T., Klein, L. and Bungert, T.J., 2022, A Call to Reflect on Evaluation Practices for Failure Detection in Image Classification. In The Eleventh International Conference on Learning Representations.

---

### Official Review · Reviewer_fXda · 2023-10-31

**Soundness:** 2 fair
**Presentation:** 2 fair
**Contribution:** 2 fair
**Rating:** 5
**Confidence:** 3

**Summary:**

In this work, the authors propose a lightweight uncertainty quantification method. More specifically, the approach accesses the bottleneck space and creates a codebook of centroids for the latent representation; then, depending on the distance from these centroids a hypothesis on the uncertainty estimation is formulated. This is achieved by a learning algorithm constituted of four different steps. The testing benchmark is performed on CIFAR10/100 and SVHN using a Wide ResNet 28-10.

**Strengths:**

- the problem of uncertainty estimation is certainly of great relevance for the deep learning community
- the idea of employing the bottleneck space  to perform uncertainty estimation is intriguing
- the authors will release the code for reproducibility

**Weaknesses:**

- it is unclear how lightweight the approach effectively is (compared to other approaches)
- the determination of the codebook size can pose serious challenges to the proposed approach
- the evaluation is performed on small-scale datasets
- only one model is employed for the evaluation

**Questions:**

- how is UA-IB compared/applies to recent works like [1,2,3]?
- how is it performing with other architectures, like ResNet-50 or to Vision Transformers?
- is the approach scaling to approaches having a larger number of classes (like ImageNet)? An experiment on ImageNet-1k would clear this point.

[1] Durasov, Nikita, et al. "Masksembles for uncertainty estimation." Proceedings of the IEEE/CVF Conference on Computer Vision and Pattern Recognition. 2021.
[2] Havasi, Marton, et al. "Training independent subnetworks for robust prediction." ICLR 2021
[3] Laurent, Olivier, et al. "Packed-Ensembles for Efficient Uncertainty Estimation." ICLR 2023

---

> ### Author Response · Authors · 2023-11-16
> **Response to Review**
>
> Thank you for reviewing our work and for your comments and suggestions! Please find our replies below.
>
>  1) *it is unclear how lightweight the approach effectively is (compared to other approaches)*
>
> We did not observe computation overhead for training our model compared to a single, deterministic network that is not uncertainty-aware. The parameter complexity of our model is provided in Table 2.
>
> 2) *the determination of the codebook size can pose serious challenges to the proposed approach*
>
> We observed that a larger-than-needed codebook will not harm the performance. We **provided some guidelines** on the selection of the codebook size right above Table 10  **in the Appendix**.
>
> 3) *only one model is employed for the evaluation*
>
> We **extended our experiments to also test our model on regression tasks in Appendix D.1**. This set of experiments broadens the impact of our paper, especially since regression problems are often overlooked by current research. In contrast to many related works, our method does not apply only to images and/ or classification tasks.
>
> 4) *the evaluation is performed on small-scale datasets*
>
> We would like to highlight that OOD tasks on small-scale datasets are not necessarily considered easier tasks. We refer you to [1], [2]. We briefly quote here: “CIFAR Benchmark is NOT Easier than ImageNet Benchmark “ [1]. Moreover, **competitive methods such as SNGP [3] or DUQ [2] follow an experimental setup identical to ours**. They were scaled/ adapted later on to the applications of interest.
>
> 5) *how is UA-IB compared/applies to recent works like [1,2,3]?*
>
> Our work belongs to the class of single-network, single-forward pass models for uncertainty quantification. The work in [4] is not a single-forward pass method (it relies on MC-Droupout). The works [5, 6] are single-forward pass, multiple-network (merged into one) methods that still suffer from high memory footprint. More importantly, ensemble methods are not distance-aware, i.e., they do not explicitly capture the distance of an unseen input from the training data points.
>
> For a quantitative comparison of our model with these works, please see our comments below:
>
> Our **model** significantly **outperforms packed-ensembles** [6]. Please see Table 1, WR Network in [6], 98.1 (AUPR) vs ours 99.4 and 96.5 (AUROC) vs ours 98.6
>
> **Our model outperforms masked-ensembles** [4]. Please see Table 1 in [4], 0.96 vs ours 99.4 (AUPR) and,  94  (AUROC) vs  ours 98.6
>
> References
>
> [1] Yang J, Wang P, Zou D, Zhou Z, Ding K, Peng W, Wang H, Chen G, Li B, Sun Y, Du X. Openood: Benchmarking generalized out-of-distribution detection. Advances in Neural Information Processing Systems. 2022 Dec 6;35:32598-611.
>
> [2] Van Amersfoort J, Smith L, Teh YW, Gal Y. Uncertainty estimation using a single deep deterministic neural network. InInternational conference on machine learning 2020 Nov 21 (pp. 9690-9700). PMLR.
>
> [3] Liu J, Lin Z, Padhy S, Tran D, Bedrax Weiss T, Lakshminarayanan B. Simple and principled uncertainty estimation with deterministic deep learning via distance awareness. Advances in Neural Information Processing Systems. 2020;33:7498-512.
>
> [4] Durasov, Nikita, et al. "Masksembles for uncertainty estimation." Proceedings of the IEEE/CVF Conference on Computer Vision and Pattern Recognition. 2021.
>
> [5] Havasi, Marton, et al. "Training independent subnetworks for robust prediction." ICLR 2021
>
> [6] Laurent, Olivier, et al. "Packed-Ensembles for Efficient Uncertainty Estimation." ICLR 2023

---

> > ### Comment · Reviewer_fXda · 2023-11-22
> > **Response to the rebuttal**
> >
> > Dear author(s)
> >
> > Many thanks for your rebuttal.
> >
> > 1. Thanks for the clarification on the computational cost. Evidently, "lightweight" has many interpretations, it is important to be more precise, and to focus on the number of trainable parameters if that is the message. As you claim that the method is "lightweight", I am expecting it to consume less memory (but this gain seems marginal compared to other approaches) or computationally wise (and you say there is no gap). Hence, it seems the approach not to be much lightweight compared to other approaches.
> >
> > 2. Thank you for indicating the appendix. It is clear that increasing k the performance improves too. However, as feared, k=10 as a trade-off is studied on CIFAR, and is not guaranteed to generalize across multiple datasets. For this an optimization over this hyper-parameter is necessary.
> >
> > 3. On this point, I can not find the details of the different architecture employed for the newly proposed experiments. If the architecture is still Wide ResNet, the weakness remains.
> >
> > 4. There are in the literature works employing ImageNet as a medium-scale dataset to experiment. Among The ones cited, I can reference for example [6].
> >
> > 5. Thank you, I am satisfied with this answer, and I invite the authors to include an extended version of this comparison in the main paper.

---

> ### Author Response · Authors · 2023-11-22
> **response to reviewer fXda**
>
> Thank you for your reply! Please find our responses below:
>
> 1) As you claim that the method is "lightweight"
>
> We used the term lightweight to differentiate our model from broad and expensive model categories such as ensembles or variational Bayesian methods and classify our model as a Deterministic Uncertainty Method (DUM), i.e., a single-forward pass and single model method. Compared to ensembles, our method is more efficient in terms of both compute time and parameters. Compared to variational methods, our method is more efficient in terms of compute time since these methods require multiple forward passes to provide uncertainty estimates. Under the DUM category, all models are considered lightweight.
>
> 2)  It is clear that increasing k the performance improves too.
>
> Please note that for cifar-10, doubling k does not provide further gains. On Imagenet, we expect a sufficient codebook size significantly smaller than the number of classes since it exhibits a coarse class hierarchy. You are right that the codebook size will need to be optimized especially when training on small memory gpus (< 32GB) since it can create a trade-off with a larger batch size.
>
> 3)  I can not find the details of the different architecture
>
> We have now added the details in section D.2.3. We note here that this is a non-image task therefore resnet architectures are not applicable.
>
> 4) here are in the literature works employing ImageNet as a medium-scale dataset to experiment
>
> We remark here that this is very subjective since it highly depends on the availability of computing resources. The authors in [6] mention: "except for ImageNet, for which we use 2 to 8 Nvidia A100-80GB". According to https://lambdalabs.com/service/gpu-cloud assuming training takes roughly a day, training a single model could cost as much as ~300$.
>
> 5)  I invite the authors to include an extended version of this comparison in the main paper
>
> We have now updated Table 1 to include Masked Ensembles in the comparisons. We will repeat the experiments ourselves to properly fill-in the stds and performance on cifar-100 tasks in the next version of the paper.

---

> > ### Author Response · Authors · 2023-11-23
> > **prelimnary results on ImageNet/ ResNet 50**
> >
> > Dear reviewer,
> >
> > We have now finished the first set of experiments on ImageNet on ResNet50. Although, we don't have OOD results yet, and we consider the experiments preliminary here are some conclusions that can shed light on the behavior of our model on large-scale, ImageNet data:
> >
> > 1) UA-IB is able to match the accuracy of 76% of a single ResNet 50 deterministic network (we compare against the pretrained network provided by Tensorflow). We think this is an important result given the compression of the network due to the UA-IB regularization (codebook is of size 30, therefore the regularization of the network is strong). A similar conclusion was reached in the VIB paper [1]. We use encoders of 128-dimension (we assume diagonal covariance in contrast to the cifar-10 experiments)
> >
> > 2)Training takes 20 minutes per epoch on 8, 48 GB A6000 GPUs and batch size 128 per core.
> >
> > 3) We did not observe any stability, convergence or non-scalability issues.
> >
> > 4) With a codebook of size 30, calibration auroc reaches 0.78. with a codebook of size 80, calibration auroc is 0.80 (this experiment is ongoing)
> >
> > We maintain the beta and tau hyperparameters similar to those of cifar-10 experiments. The full details of the backbone architecture and the hyperparameters not relevant to UA-IB can be found here:
> >
> > https://github.com/google/uncertainty-baselines/blob/main/baselines/imagenet/deterministic.py
> >
> > Beyond the codebook size, we have not further fine-tuned yet.
> >
> > [1] Alemi AA, Fischer I, Dillon JV, Murphy K. Deep variational information bottleneck. arXiv preprint arXiv:1612.00410. 2016 Dec 1.

---

### Official Review · Reviewer_rmmd · 2023-11-02

**Soundness:** 2 fair
**Presentation:** 2 fair
**Contribution:** 1 poor
**Rating:** 1
**Confidence:** 5

**Summary:**

The authors propose to compress the dataset using a codebook and compute the distances from the codebook. The idea is to turn the training points through parametric distributions. Those are compressed in order to identify the centroids. The expected distance from the centroids is then used to estimate the uncertainty on the test datapoints. The authors show a few experiments in out of distribution detection and misclassification detection.

**Strengths:**

- Despite the method induces a significant drop in accuracy, it seems to produce better AUROC for misclassification detection (which is interesting).
- The conceptual aspect of the paper looks theoretically reasonable and is validated through a simple toy experiment ... (see weakness)

**Weaknesses:**

Factual errors:
- Classifying Guassian Process Models like SNGP as deterministic is not correct.
- The fact DUMs that apply regularisation and obtain SOTA OOD detection need to harm calibration is false. Works like RegMixup [1], AugMix [2] and PixMix [3] clearly show this is not the case.

Weaknesses:
- ... however good performance on toy experiments does not necessarily reflect in useful uncertainties in real datasets, and viceversa.
- There are several deterministic techniques that induces better uncertainty estimation, that the authors neglect. Besides the already mentioned RegMixup [1], AugMix [2] and PixMix [3]. These techniques add no parameters to the model. Furthermore the authors should consider other fundamental baselines from [0].
- The paper trains on a single dataset, CIFAR-10, which is extremely simple. While some of the baselines the authors selected are known not to scale beyond that (e.g. DUQ, which shares a few conceptual similarities to the proposed work, becomes unstable when the number of classes increases), others (SNGP, Deep Ensemble) do scale and therefore the authors should show extensive comparisons on larger scale datasets (CIFAR-100, ImageNet are the bare-minimum; when evaluating on ImageNet-O, please be mindful about the caveat indicated in [4] about the data imbalance).
- Especially compared to the deterministic baselines mentioned above, the method induces an accuracy drop.

[0] https://arxiv.org/pdf/2210.07242.pdf
[1] https://arxiv.org/abs/2206.14502
[2] https://arxiv.org/abs/1912.02781
[3] https://arxiv.org/abs/2112.05135
[4] https://arxiv.org/pdf/2207.11347.pdf

**Questions:**

- Could the authors provide more extensive comparisons with state-of-the-art baselines?
- Could the authors prove their method scales to datasets of increasing complexity? This is especially related to understanding whether convergence can be achieved for larger codebook size that is inevitably required for more complex datasets.
- Could the authors show how the model behaves for distribution shift? The drop in accuracy on in-distribution test-sets is concerning.
- Given newer and more powerful architectures exist, could the authors test the validity of their method on models like ConvNeXt, ViT etc.? As these models are becoming more and more relevant than WideResNet28-10 (which is, by the way, extremely overparametrised for the task at hand).

---

> ### Author Response · Authors · 2023-11-16
> **Response to Review**
>
> Thanks for all the feedback! Please find our comments and replies below:
>
>
> 1) *Classifying Guassian Process Models like SNGP as deterministic is not correct.*
>
> **SNGP is a probabilistic model with deterministic uncertainty estimates.** This is because estimating the likelihood of the model is a deterministic function of the data. We borrowed terminology used in [1] (for example, **in Table 1 in [1], SNGP is listed as a DUM**). We agree that this can be confusing and that maybe a term such as single-forward pass uncertainty method would be more appropriate.
>
> 2) *The fact DUMs that apply regularisation and obtain SOTA OOD detection need to harm calibration is false. Works like RegMixup [1], AugMix [2] and PixMix [3] clearly show this is not the case*.
>
> In the main text, DUMs, as in [1], refer only to Gaussian Process models and clustering methods that do harm calibration [1]. **For completeness and clarity, we now discuss dataset augmentation methods in Section 6.**
>
> 3) *Especially compared to the deterministic baselines mentioned above, the method induces an accuracy drop.*
>
> **Our method does not induce an accuracy drop.** We refer you to Table 2: It performs on par with other DUMs (95.9% accuracy) that do not use dataset augmentations.
>
> 4) *There are several deterministic techniques that induces better uncertainty estimation, that the authors neglect. Besides the already mentioned RegMixup [1], AugMix [2] and PixMix [3]. These techniques add no parameters to the model. Furthermore the authors should consider other fundamental baselines from [0]*
>
> **We have revised our results (Table 1) to add two recent baselines.** For example, compared to the suggested RegMixup baseline, our method achieves stronger OOD performance: 96% vs 98% (AUROC) for SVHN and 89 vs 92% (AUROC) for CIFAR-100.
>
> 5) *The paper trains on a single dataset, CIFAR-10, which is extremely simple.*
>
> Although **CIFAR-10 ** is a smaller-scale problem, it is widely regarded as a **challenging OOD task not easier than ImageNet.** [2], [3]. **In Appendix D.1 we added an additional experiment to test our model on two additional real datasets for regression tasks.** Several baselines in current research treat only images and/or classification tasks. Our method provides a unified uncertainty score for both tasks.  UA-IB performs better or on par with an ensemble of 4 networks on all datasets and all tasks.
>
> 6) *DUQ, which shares a few conceptual similarities to the proposed work, becomes unstable when the number of classes increases*
>
> **We do not expect that UA-IB will not scale due to the similarities to DUQ.** Some key differences:
>
>  a. DUQ is hardwired to centroids that must be equal to the number of classes. Our model, in contrast, supports arbitrary codebook sizes that do not necessarily match the number of classes (please see Table 10 in Appendix).
>
> b. Our model can handle regression tasks as well (in contrast to DUQ).
>
> c. In our method, we use a different loss function where the assignment of datapoints to centroids becomes in an unsupervised manner.
>
>
> 7) *(SNGP, Deep Ensemble) do scale*
>
> Fundamental research and empirical research/ scaling experiments are both important. The focus of this work lies in the first category and is to provide a fresh analysis of UQ as a lossy compression of the training dataset. Please also note that **other baseline DUMS (that do not use augmentations) do not provide performance on Imagenet OOD tasks.** Finally, the impactful SNGP was scaled on ImageNet recently, after its publication as an extension of prior work.
>
> References:
>
> [1] Postels J, Segu M, Sun T, Sieber L, Van Gool L, Yu F, Tombari F. On the practicality of deterministic epistemic uncertainty. arXiv preprint arXiv:2107.00649. 2021 Jul 1.
>
> [2] Yang J, Wang P, Zou D, Zhou Z, Ding K, Peng W, Wang H, Chen G, Li B, Sun Y, Du X. Openood: Benchmarking generalized out-of-distribution detection. Advances in Neural Information Processing Systems. 2022 Dec 6;35:32598-611.
>
> [3] Van Amersfoort J, Smith L, Teh YW, Gal Y. Uncertainty estimation using a single deep deterministic neural network. InInternational conference on machine learning 2020 Nov 21 (pp. 9690-9700). PMLR.

---

> > ### Comment · Reviewer_rmmd · 2023-11-16
> > **Thanks for the rebuttal, my concerns about experiments remain unchanged.**
> >
> > Thank you for the response, I have also checked the modifications made in the paper. I find the empirical side to be the weakest and most problematic aspect of the paper.
> >
> > 1 -  Fine, the point is arguable but minor.
> >
> > 2- Can the authors also compare with AugMix and PixMix? Furthermore the numbers of Packed Ensemble and RegMixup seem to be copy-pasted from the respective papers. This is not an acceptable comparison. For instance, the Deterministic (DNN) in the RegMixup paper has lower OOD performance with respect to the one reported by the authors. The authors should run both Packed Ensemble and RegMixup themselves, with the same architecture implementation and training loop and tune the hyperparameters to ensure a fair comparison, as all these factors can affect performance.
> >
> > 3 - Agreed, my mistake. I would invite the authors to consider more recent baselines from [0] still.
> >
> > 4 - I kindly ask again the authors to expand their experimental setting by choosing multiple OOD tasks with varying levels of difficulty, e.g. [0] could be a reference to find hard tasks. Whether the model scales or not is to be verified empirically and cannot be conjectured. This is the strongest limitation of the paper.
> >
> > 4.1 -  As for the UCI datasets, can the authors specify exactly what model is being trained? If it's a simple MLP, given the low dimensionality and simplicity of the data and the models, the computational cost of training 4 members of an ensemble is totally negligible and produces superior or comparable performance.
> >
> > 5 - SNGP has not been scaled "recently" to ImageNet, but soon after the code was published. I do not require to perform ImageNet-O experiments necessarily, but I ask to use datasets representative of real data (not 32x32 images) and real out-of-distribution tasks with a high amount of classes and complexity. For CIFAR-10, the simple fact the AUROC of the baseline is around 95% and 89%  (out of 100%) tells it is not a "hard" task. The value for the baseline can go much lower in harder settings.
> >
> > 6 - If the augmentation baselines are truly composable with your method is to be verified empirically.

---

> > > ### Author Response · Authors · 2023-11-22
> > > **second response to reviewer**
> > >
> > > Thank you for your reply. Please find our comments below:
> > >
> > > 1) The authors should run both Packed Ensemble and RegMixup themselves,
> > >
> > > We agree that a more rigorous analysis is needed. We will repeat the experiments to take care of the hyperparameter tuning and fill in the table with stds and the auprc score in the next version of the paper. However, we consider this point minor: 1) Packed Ensemble is not a direct competitor of our method since not a DUM 2) RegMix significantly underperforms many of the baselines. Given that we used the numbers reported on the wideResNet architecture, it would be concerning for the RegMix method if it exhibited such a sensitivity to the hyperparameters.
> > >
> > > 2) the simple fact the AUROC of the baseline is around 95% and 89% (out of 100%) tells it is not a "hard" task
> > >
> > > Please note that AUROC is not always a good indicator of the difficulty of the task. This is because it is sensitive to class imbalance. In fact, this is the case for the ImageNet-O dataset (2000 images) which is much smaller than ImageNet's test set (50000 images). We find AUPRC more informative in these settings.
> > >
> > > 3) As for the UCI datasets, can the authors specify exactly what model is being trained
> > >
> > > The main purpose of this experiment was to demonstrate the consistent OOD capability of UA-IB beyond images or classification tasks.
> > >
> > > 4) SNGP has not been scaled "recently" to ImageNet, but soon after the code was published
> > >
> > > We refer you to the [journal version of SNGP](https://www.jmlr.org/papers/volume24/22-0479/22-0479.pdf) which reports the ImageNet results:  The paper was submitted for publication almost two years after the NeurIPS paper. Unfortunately, we cannot know exactly when the experiments have been finalized unless they have been officially disseminated.

---

> > > > ### Comment · Reviewer_rmmd · 2023-11-22
> > > > **Empirical evidence not provided, concerns dodging unacceptable**
> > > >
> > > > 1) We cannot establish anything until the rigorous analysis is run. It's not simply about hyperparameters when the two cross-entropy trained networks in the two papers differ so much, indicating a fundamental difference in the setup of the two codebases that goes beyond hyperparameters choice. (For instance, I also understand the authors have been using tensorflow: the same models trained in tensorflow or pytorch can behave differently in a way that affects the considered metrics even when the methodology is the same). I invite the authors to carry out fair experiments to produce factual evidence and share the code of the implementation of all the baselines to guarantee the correctness of implementation. I also invite the authors, for the third time, to compare with AugMix and PixMix (in a fair setting, as it is expected for serious scientific investigation).
> > > >
> > > > 2)  I invite the authors to run their code evaluating the AUROC by flipping the choice of the positive and negative class in presence of strong class imbalance (e.g. on C10 vs SVHN, though the imbalance is not as strong as on ImageNet-O vs ImageNet ... experiment which the authors keep refusing to run for mysterious reasons). To their surprise, the authors will see the AUROC is stable in the presence of class imbalance, while the AUPRC is not. This is mathematically provable in the specific setting of OOD detection. Whether the AUROC or AUPRC are stable to class imbalance remains meaningless to the discussion. It looks more like an attempt to cherry-pick which metric is more convenient to defend the lack of willingness to run proper experiments on multiple datasets.
> > > >
> > > > 3) Well, it either does comparatively or worse for some metrics than an ensemble of linear regressors (I have to guess, given the lack of response), which don't take much to train even on CPU. It doesn't prove to be much more useful. If the authors want to prove their method is useful in the problem of regression, they could consider more complex regression tasks that require training proper neural networks and cannot be performed with ensembles (or for which training ensembles would be significantly more expensive than using their method).
> > > >
> > > > 4) The authors could check the github history of the benchmark with which SNGP was published to see when SNGP was precisely extended to ImageNet, long before the journal version of the paper. It is anyways irrelevant, as the statement I had made was related to the fact the authors were using the date in which SNGP was supposed to be extended to ImageNet as an excuse not to perform further experiments. The real concern I had raised at point 5 is pretty clear.
> > > >
> > > > As a general comment, I find the lack of willingness to run proper and extensive experiments extremely concerning about the scalability and validity of the method. I also find the attempt to dodge the main points of the criticism by focusing on minor aspects disrespectful of my time. I start doubting the honesty of the authors. I will not further discuss unless extensive and adequate empirical evidence is provided. Citing papers and extrapolating sentences is not a replacement for empirical evidence.

---

> ### Author Response · Authors · 2023-11-23
> **preliminary results on ImageNet**
>
> Thanks for all the time you have put into giving feedback on the paper and clarifying the concerns.
>
> 1) On the ImageNet experiments
>
>  We have started running the experiments on ImageNet. We haven’t worked on the AugMix and PixMix yet since we were dedicating our efforts and resources to the scalability concerns you raised (which we also think are the most important) and need to be prioritized given the tight timeframe and the resources we currently have. While still very preliminary results,  here are some conclusions we can draw.
>
> With a codebook of size 30 and encoders of 128-dimension (we assume diagonal covariance in contrast to the cifar-10 experiments), we observe the following:
>
> * UA-IB is able to match the accuracy of 76% of a single ResNet 50 deterministic network (we compare against the pretrained network provided by Tensorflow). We think this is an important result given the compression of the network due to the UA-IB regularization. A similar conclusion was reached in the VIB paper [1].
>
> * Training takes 20 minutes per epoch on 8, 48 GB A6000 GPUs and batch size 128 per core.
>
> * We did not observe any stability or non-scalability issues.
>
> * Calibration AUROC reaches 0.78.
>
> Edit: we started a second experiment right after posting this with codebook size 80. The Calibration AUROC is currently 0.80 (training has not finished yet).
>
> We do not have results on OOD experiments yet. Moreover, we have not finetuned the codebook size and the rest of the hyperparameters yet. This result refers to a single configuration we tried and took 1.5 days to complete given the hardware we currently have ( a single 8-GPU server).
>
> 2) On the regression experiments
>
>  *It either does comparatively or worse for some metrics than an ensemble of linear regressors (I have to guess, given the lack of response)*
>
> We definitely expect that large ensembles will outperform UA-IB as well as other DUMs. However, **we do not consider ensembles direct competitors of DUMs**.
>
> [1] Alemi AA, Fischer I, Dillon JV, Murphy K. Deep variational information bottleneck. arXiv preprint arXiv:1612.00410. 2016 Dec 1.

---

> ### Author Response · Authors · 2023-11-23
> **minor correction on our class imbalanced claim**
>
> Albeit not critical, we agree with your claim:
>
> "I invite the authors to run their code evaluating the AUROC by flipping the choice of the positive and negative class in the presence of strong class imbalance "
>
> but one has to be careful to also flip the score appropriately.
>
> However, we insist on the difficulty/ importance of the cifar-10 benchmarks. We refer you to empirical evidence provided by other researchers, particularly in Table 1 of: https://arxiv.org/pdf/2210.07242.pdf
>
> In Table 1, you will see that methods on **Far-OOD tasks on Imagenet achieve higher scores** compared to the far-ood tasks for the cifar datasets. This conclusion is made clear in the main text:
>
> **CIFAR Benchmark is NOT Easier than ImageNet Benchmark We find that the OOD detection
> performance score for the ImageNet dataset is generally higher than that of CIFAR-10 and CIFAR100, which is another surprising discovery considering ImageNet is composed of more complex data than others and seems difficullt**

---

### Official Review · Reviewer_81b3 · 2023-11-09

**Soundness:** 3 good
**Presentation:** 3 good
**Contribution:** 2 fair
**Rating:** 8
**Confidence:** 3

**Summary:**

the paper introduces the uncertainty aware information bottleneck (ua-ib), an approach to quantify uncertainty in machine learning. the ua-ib is positioned as a method that integrates uncertainty estimation directly into the learning process. contributions include theoretical formulation, empirical validation, and comparative analysis with existing methods.

**Strengths:**

1. in my eyes, ua-ib's primary strength lies in its innovative blend of the ib ansatz with a new practical spin. the theoretical foundation is well-established, drawing from the classical information bottleneck principle and extending it in a novel direction.
1. the paper goes beyond theory, providing empirical evidence that the ua-ib framework can be operationalized. the method's capability to quantify uncertainty is assessed through a series of experiments, ranging from noise-infused synthetic datasets to real-world data.
1. ua-ib's algorithmic formulation appears efficient and scalable within the bounds of the proposed experiments. the authors' approach to model complexity, as detailed in section 4, suggests some consideration for practical constraints.
1. the robustness to data anomalies, explored in section 5.2, hints at the method's potential to deliver when faced with data deviating from the training distribution, an essential feature for real-world applications.

**Weaknesses:**

1. despite the comprehensive nature of the ua-ib framework, the manuscript does not adequately tackle the question of computational efficiency in real-world scenarios. while the authors provide a cursory overview of the method's computational requirements, they stop short of a full exploration, leaving the reader to speculate about ua-ib's performance in larger, more complex environments.
1. the treatment of hyperparameters, although mentioned, is insufficiently detailed. the paper would benefit from a dedicated section that delves into how hyperparameter affects the ua-ib performance.
1. the scalability of ua-ib is not convincingly demonstrated. the experiments conducted are robust, yet they do not encompass the scale of data that would be encountered in many practical applications, such as large-scale image or language processing tasks.
1. the potential for integration of ua-ib with other learning paradigms or frameworks is mentioned in passing but is not explored in depth. the ability to integrate ua-ib post-hoc with existing frameworks is critical for its adoption.
1. lightweight uq for dnns has a long and colorful history. while i appreciate the dense context provided by authors and in particular the original tie-in of tali tishby's ib concept, a range of lighweight uq methods are not mentioned. for example, gast's probout which also offers a layerwise propagation version (https://openaccess.thecvf.com/content_cvpr_2018/html/Gast_Lightweight_Probabilistic_Deep_CVPR_2018_paper.html), quantile regression (https://www.jstor.org/stable/1913643) and conformal prediction (https://www.jmlr.org/papers/volume9/shafer08a/shafer08a.pdf), interval neural networks (https://arxiv.org/abs/2003.11566) as well as the classic direct variance prediction (https://proceedings.neurips.cc/paper/1994/hash/061412e4a03c02f9902576ec55ebbe77-Abstract.html).
1. code is not provided, only after acceptance.

**Questions:**

1. could you provide a comprehensive analysis of ua-ib's computational demands, specifically addressing its performance with large-scale, complex datasets prevalent in real-world applications?

2. could you elaborate on how hyperparameter choices affect ua-ib's performance, especially considering computational constraints?

3. can you share additional experimental results or simulations that demonstrate ua-ib's scalability to the data sizes seen in high-dimensional image or language tasks?

4. could you discuss potential post-hoc integration strategies for ua-ib with trained models and the expected challenges or benefits?

5. could you situate the ua-ib's approach in the broader landscape of lightweight uq or explain why certain methods are not relevant in your view?

6. why are you not sharing the code for reviewing?

---

> ### Author Response · Authors · 2023-11-16
> **Response to Review**
>
> We thank the reviewer for the thorough and detailed review. Please find our replies below.
>
>
> 1) *the paper would benefit from a dedicated section that delves into how hyperparameter affects the ua-ib performance*.
>
> As suggested, we have **added ablation studies** over UA-IB’s hyperparameters in **Appendix D.3**  in the revised version of the paper.
>
> 2) *could you discuss potential post-hoc integration strategies for ua-ib with trained models and the expected challenges or benefits?*
>
> One way this could be done is to apply UA-IB (i.e., codebook and encoder of the pretrained features) on features extracted from the pretrained network. The original VIB paper (section 4.2.5 in [1]) considers a similar setup. **We have revised the paper to discuss this post-hoc integration in Sec. 7**.
>
> 3) *why are you not sharing the code for reviewing?*
>
> We have added the code to the supplemental materials.
>
> 4) *scalability to the data sizes seen in high-dimensional image or language tasks?*
>
> a. Fundamental methods research and scaling research are both important. This work is studying the former. For example, the impactful SNGP paper [2] was not originally demonstrated on large-scale images. However, we do agree that large-scale experiments will increase the impact of our paper.
>
> b. We are actively working on that.
>
> c. Although **CIFAR-10** is a smaller-scale problem, it is **widely regarded as a challenging OOD task as difficult as ImageNet**. [3], [4].
>
> d. To broaden the impact of our paper,  we expanded our experiments and tested our model on two regression tasks in Appendix d.1.
>
> 5) *could you situate the ua-ib's approach in the broader landscape of lightweight uq or explain why certain methods are not relevant in your view?*
>
> At a high level, one key difference between UA-IB and all the rest of UQ methods is that it both regularizes the network  (since the codebook is much smaller than the number of training datapoints) and equips it with distance-aware uncertainties (uncertainties that should grow away from the training data). Moreover, our method can measure uncertainty for both regression and classification tasks in contrast to quantile regression, interval networks, and variance prediction which handle only the former. Regarding [7], it can model only aleatoric uncertainty. The primary concern of our work, similar to Gaussian Process models, ensembles, Bayesian Neural Network etc is the estimation of epistemic uncertainty instead which is generally considered a more challenging task [5]. Finally, conformal prediction is a post-hoc uncertainty quantification method. However, leveraging uncertainty during training (or at a fine-tuning phase where UA-IB and backbone network are not jointly trained) can also help the model improve its accuracy [6]. We discuss this as future work in the draft.
>
> References:
>
> [1] Alemi AA, Fischer I, Dillon JV, Murphy K. Deep variational information bottleneck. arXiv preprint arXiv:1612.00410. 2016 Dec 1.
>
> [2] Liu, J., Lin, Z., Padhy, S., Tran, D., Bedrax Weiss, T. and Lakshminarayanan, B., 2020. Simple and principled uncertainty estimation with deterministic deep learning via distance awareness. Advances in Neural Information Processing Systems, 33, pp.7498-7512.
>
> [3]Yang J, Wang P, Zou D, Zhou Z, Ding K, Peng W, Wang H, Chen G, Li B, Sun Y, Du X. Openood: Benchmarking generalized out-of-distribution detection. Advances in Neural Information Processing Systems. 2022 Dec 6;35:32598-611.
>
> [4]Van Amersfoort J, Smith L, Teh YW, Gal Y. Uncertainty estimation using a single deep deterministic neural network. In International conference on machine learning 2020 Nov 21 (pp. 9690-9700). PMLR.
>
> [5] Kendall, A. and Gal, Y., 2017. What uncertainties do we need in bayesian deep learning for computer vision?. Advances in neural information processing systems, 30.
>
> [6] Wilson AG, Izmailov P. Bayesian deep learning and a probabilistic perspective of generalization. Advances in neural information processing systems. 2020;33:4697-708.
>
> [7] Gast J, Roth S. Lightweight probabilistic deep networks. InProceedings of the IEEE Conference on Computer Vision and Pattern Recognition 2018 (pp. 3369-3378).

---

> ### Comment · Reviewer_81b3 · 2023-11-21
> **rebuttal response**
>
> dear authors,
>
> thank you very much for your comprehensive response.
>
> i very much appreciate the detailed ablation experiments in d.3 as well as the release of the code.
>
> i raised my score to accept in response to your revisions.
>
> the experiments still leave room for improvement wrt the complexity of the data. i also think that related work should be contextualized more fully in the final version, as pointed out by me and other reviewers.
>
> however, i do not agree with other reviewers that these shortcomings invalidate the overall contribution.
>
> in good spirits,
>
> reviewer 81b3

---

### Meta-Review · Area_Chair_acQ6 · 2023-12-09

**Metareview:**

The authors aim at developing a lightweight uncertainty quantification method as an alternaive to more costly methods such as deep network ensembles. Their approach augments prior information bottleneck methods with a codebook in order to obtain a compressed representation of all training inputs. Given a new example, uncertainty is then quantified in terms of its distance from the codebook.

Although the reviewers raise a number of critical points in their original reports, there is agreement that the paper has a high potential, and the authors idea looks quite intriguing. The authors also showed a high level of commitment during the rebuttal phase and did their best to respond to the comments and to improve the submission. This was appreciated and positively acknowledged by all. In the discussion between authors and reviewers, some critical points could be resolved and some questions clarified. Other points remained open and were critically reconsidered in the subsequent internal discussion. In particular, the experimental part was still found to be lacking essential components to make the paper convincing and impactful. Moreover, the effectiveness of the method is still difficult to judged. Eventually, the submission was found to remain a bit behind the expectations for a top venue such as ICLR.

**Justification For Why Not Higher Score:**

Open questions regarding scalability of the method and effectiveness. Experimental part is not convincing.

**Justification For Why Not Lower Score:**

N/A

---

### Decision · Program_Chairs · 2024-01-16

Reject